# SmartThinker: Progressive Chain-of-Thought Length Calibration for Efficient Large Language Model Reasoning

**Chenzhi Hu** [1]  **Qinzhe Hu** [1]  **Yuhang Xu** [1]  **Junyi Chen** [1]
**Ruijie Wang** [2]  **Shengzhong Liu** [1]  **Jianxin Li** [2]  **Fan Wu** [1]  **Guihai Chen** [1]

## Abstract

Large reasoning models (LRMs) like OpenAI o1 and DeepSeek-R1 achieve high accuracy on complex tasks by adopting long chain-of-thought (CoT) reasoning paths. However, the inherent verbosity of these processes frequently results in redundancy and overthinking. To address this issue, existing works leverage Group Relative Policy Optimization (GRPO) to reduce LRM output length, but their static length-reward designs fail to adapt to problem difficulty and response-length distributions, causing over-compression and compromised accuracy. Therefore, we propose *SmartThinker*, a novel GRPO-based efficient reasoning method with progressive CoT length calibration. *SmartThinker* makes a two-fold contribution: First, it dynamically estimates the optimal length with peak accuracy during training and guides overlong responses toward it to reduce reasoning length while sustaining accuracy. Second, it dynamically modulates the length-reward coefficient to avoid the unwarranted penalization of correct reasoning paths. Extensive experimental results show that *SmartThinker* achieves up to 52.6% length compression with improved accuracy and achieves up to 16.6% accuracy relative improvement on challenging benchmarks like AIME25. The source code can be found at https://github.com/SJTU-RTEAS/SmartThinker.

## 1. Introduction

The evolution of Large Language Models (LLMs) is currently undergoing a paradigm shift from naive pattern match-

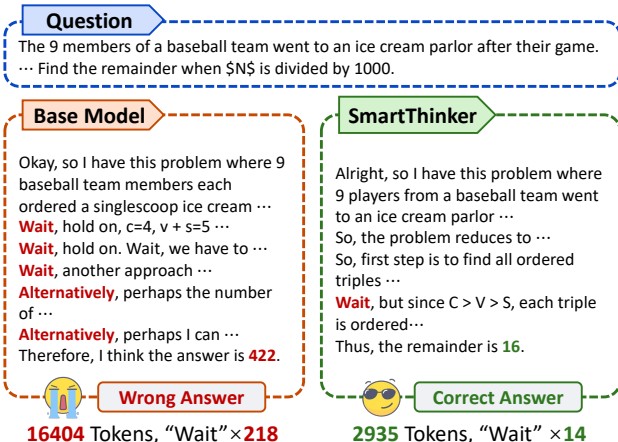

*Figure 1.* An illustrative comparison between the base model and *SmartThinker* on an AIME25 problem.

ing (Chen et al., 2025) to deliberate logical reasoning. While traditional models excel at providing rapid, associative responses, they often struggle with the multi-step rigor required for complex mathematics, coding, and scientific discovery. This limitation has catalyzed the rise of Large Reasoning Models (LRMs) like OpenAI o1 (Jaech et al., 2024) and DeepSeek-R1 (Guo et al., 2025). Unlike their predecessors, relying primarily on the pre-training scale, LRMs leverage inference-time scaling laws. By integrating reinforcement learning (RL) with Chain-of-Thought (CoT) (Wei et al., 2022) processing, LRMs like DeepSeek-R1 (Guo et al., 2025) achieve deep thinking, allocating more computational power during the generation phase to verify, correct, and refine their own logic.

However, the reliance on long reasoning traces induces a fundamental challenge known as the *overthinking problem*. While increasing reasoning length facilitates answering difficult problems, excessively long chains of thought often lead to diminishing or even negative returns. On one hand, overthinking consumes excessive tokens, leading to unnecessary computational and time overhead (Chen et al., 2026; Xu et al., 2026b); on the other hand, overthinking simple problems may cause the model to randomly diverge and miss the correct answer. These observations indicate **the reasoning length should be carefully controlled to bal-**

[1]Shanghai Jiao Tong University [2]Beihang University. Correspondence to: Shengzhong Liu <shengzhong@sjtu.edu.cn>, Fan Wu <wu-fan@sjtu.edu.cn>.

*Proceedings of the 43rd International Conference on Machine Learning*, Seoul, South Korea. PMLR 306, 2026. Copyright 2026 by the author(s).

**ance correctness and efficiency, rather than maximized indiscriminately**.

Motivated by this observation, recent studies have explored strategies for efficient LLM reasoning. These methods can be broadly categorized into 1) *training-free approaches* (Xu et al., 2026a; Lin et al., 2026), which improve efficiency at inference or prompt level without updating model parameters, and 2) *training-based approaches* (Aggarwal & Welleck, 2025; Hou et al., 2025; Yi et al., 2026; Tu et al., 2026), which explicitly shape the model's reasoning behavior through optimization objectives such as supervised fine-tuning and reinforcement learning with length-aware reward designs. Although training-free methods are flexible and computationally inexpensive, training-based approaches, particularly reinforcement learning, tend to achieve more consistent efficiency-accuracy gains by directly optimizing reasoning trajectories. Particularly, Group Relative Policy Optimization (GRPO) (Shao et al., 2024) improves the sample efficiency and training stability of reinforcement learning and has become a common basis for efficient reasoning method design.

Most GRPO-based mechanisms incorporate a length reward to encourage shorter reasoning trajectories and assign a higher advantage to outputs with fewer tokens. While they are effective in compressing Chain-of-Thought and sometimes even improve accuracy, they rely heavily on heuristic assumptions to estimate the optimal reasoning length. In particular, the target length is not explicitly modeled according to correctness, causing linear length penalties to deviate from the true length–accuracy tradeoff, and often overshoot the optimal reasoning length needed for problem solving. Moreover, existing length reward (Aggarwal & Welleck, 2025; Tu et al., 2026; Yi et al., 2026) designs are usually static and task-agnostic, applying the same length reward function across problems of varying difficulty. They fail to account for the fact that harder problems inherently require longer and more exploratory reasoning, leading long but correct trajectories to be penalized similarly to incorrect ones. As a result, such static formulations sacrifice necessary reasoning diversity and inevitably degrade response quality to complex questions.

To bridge this gap, we propose *SmartThinker*, a GRPO-based algorithm that **jointly optimizes reasoning accuracy and efficiency through adaptive length calibration with dynamic length reward based on optimal length estimation**. Our approach is "smart" in two key aspects. First, instead of heuristically penalizing reasoning length, we explicitly characterize the relation between reasoning length and correctness using a Gaussian distribution, enabling us to identify an optimal reasoning length maximizing the probability of success for a given prompt. This replaces blind linear penalties with a principled probabilistic objective.

Second, we introduce a dynamic length-reward coefficient that ensures the normalized advantage of correct trajectories remains non-negative, preventing valid but longer reasoning paths from being mistakenly suppressed while skipping manual hyperparameter tuning.

We extensively evaluate *SmartThinker* on base models of different scales and mathematical benchmarks of varying difficulty. Experimental results show that *SmartThinker* reduces reasoning token usage by up to 52.6% while improving reasoning accuracy. On challenging benchmarks like AIME25, accuracy gains a relative improvement of up to 16.6%, demonstrating the effectiveness of adaptive reasoning length. Figure 1 presents an illustrative example from AIME25.

Our contributions are summarized as follows:

- We identify and analyze the reward design issues in GRPO-based efficient reasoning methods caused by the lack of dynamic reward design.

- We propose a probabilistic approach to estimate the optimal reasoning length for each question and design a corresponding dynamic length reward.

- We use a dynamic length-reward coefficient to calibrate the weight of the length reward in the total reward, avoiding incorrectly penalizing correct trajectories.

- Extensive experiments demonstrate that *SmartThinker* can simultaneously improve both efficiency and accuracy with length reduction of up to 52.6% and accuracy improvement of up to 16.6%.

## 2. Background and Motivation

### 2.1. Overthinking Phenomenon in LLM Reasoning

A central motivation of efficient reasoning is the *overthinking* phenomenon, where models expend excessive computational effort on relatively simple tasks, generating unnecessarily long and convoluted reasoning chains that exceed the actual problem complexity. This behavior leads to both inefficiency and increased risk of reasoning errors. From the perspective of reasoning length, overthinking manifests as excessive *verbosity and computational overhead* in Chain-of-Thought prompting (Han et al., 2025), as well as *performance degradation* caused by overly long reasoning traces (Chen et al., 2024).

Recent studies further show that reasoning accuracy does not monotonically improve with longer outputs. Instead, the relationship between reasoning length and accuracy typically follows an inverted U-shaped curve, with performance peaking at an intermediate, globally optimal length (Wu

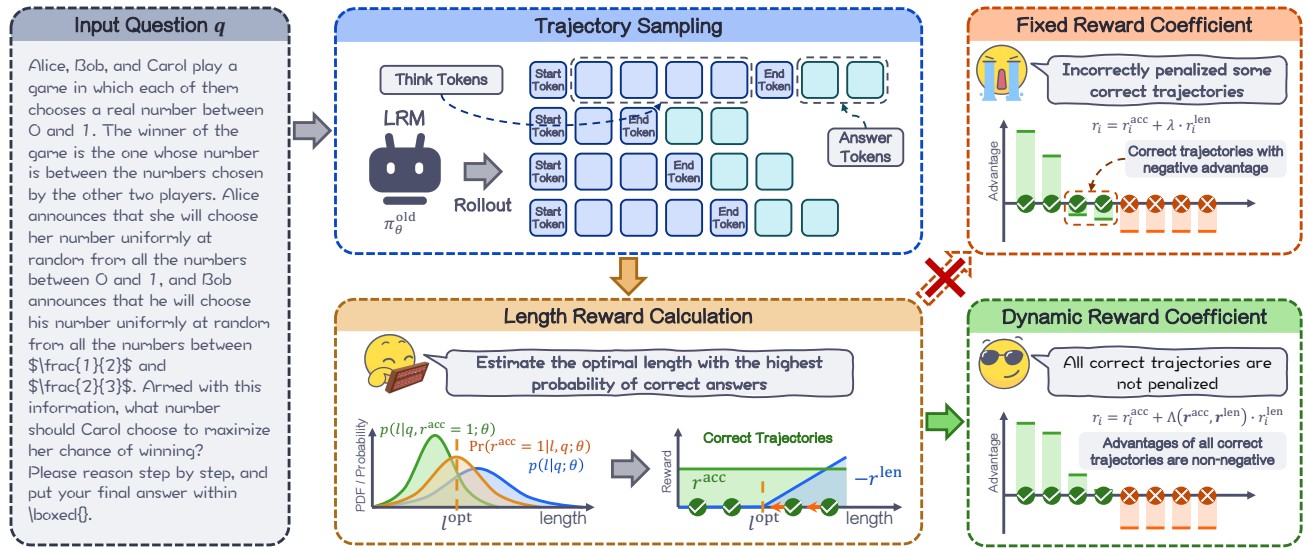

*Figure 2.* Overview of *SmartThinker*'s dynamic reward and advantage calculation process.

et al., 2025). Our experimental results, as shown in Figure 3, also demonstrate that extending the reasoning beyond this optimal point can even harm accuracy. These observations motivate approaches that explicitly guide models toward an appropriate reasoning length, rather than blindly encouraging longer or shorter reasoning paths.

## 2.2. GRPO for Efficient Reasoning

To address efficiency issues in LLM reasoning, reinforcement learning (RL) has been widely adopted in post-training of large reasoning models (LRMs). Among existing methods, Group Relative Policy Optimization (GRPO) (Shao et al., 2024) has emerged as a popular and effective approach.

As a prerequisite, we first briefly introduce GRPO algorithm. As a novel LLM reinforcement learning (RL) method, GRPO has been widely used in post-training of large reasoning models (LRMs).

We provide a complete version of the GRPO algorithm in Appendix A.3. Here we discuss its simplified form. In each training step, the old policy $\pi_{\text{old}}$ will generate a group of trajectories. Given question $q$, for each trajectory $o_i$ in a group, a simplified training objective can be written as

$$\max_{\theta} \frac{\pi_{\theta}\left(o_i \mid q\right)}{\pi_{\text{old}}\left(o_i \mid q\right)} \hat{A}_i, \tag{1}$$

where $\pi_{\theta}$ is the current policy to be updated and $\theta$ is its weights. the normalized advantage $\hat{A}_i$ is calculated as

$$\hat{A}_i = \frac{r_i - \text{mean}(\mathcal{R})}{\text{std}(\mathcal{R})}, \tag{2}$$

where $\mathcal{R} = \{r_1, \ldots, r_G\}$ is the set of rewards within a group. When $A_i > 0$, the GRPO algorithm tends to increase

the maximum likelihood of $(o_i \mid q)$. Conversely, when $A_i < 0$, the GRPO algorithm tends to decrease the maximum likelihood of $(o_i \mid q)$.

Most existing GRPO-based approaches for efficient reasoning rely on a fixed reward $r_i$ formulation that combines accuracy and length:

$$r_i = r_i^{\text{acc}} + \lambda r_i^{\text{len}}, \tag{3}$$

where $r_i^{\text{acc}}$ and $r_i^{\text{len}}$ represents the accuracy and length reward respectively. Generally speaking, $r^{\text{len}}$ is monotonically non-increasing with respect to $l_i$ to reward shorter trajectories, and $\lambda$ is a coefficient.

## 2.3. Limitations of Static GRPO Length Reward

It is worth noting that the reward formulation in Eq. (3), while simple and effective in encouraging shorter reasoning, exhibits several fundamental limitations when applied to efficient reasoning. In particular, the design is *static* in nature, which manifests in two key aspects:

**1) Static length reward.** In Eq. (3), the length reward $r_i^{\text{len}}$ is computed solely based on the length of the individual trajectory $o_i$, independent of other trajectories sampled for the same prompt. Consequently, the reward fails to account for the joint distribution of length and correctness within a GRPO group, which implicitly reflects the relative difficulty of the input question under the current model. Although some methods (Yi et al., 2026; Liu et al., 2025b) design different rewards for varying difficulties, they only consider a few discrete cases separately, lacking continuous control over difficulty.

**2) Static length-reward coefficient.** Although some methods consider the first case, dynamically discussing length rewards based on in-group accuracy, the reward formula-

tion in Eq. (3) itself remains static in another crucial aspect. Specifically, Eq. (3) employs a fixed coefficient $\lambda$ to balance accuracy and length rewards across all trajectories. Even when the length reward is adapted using group-level statistics, a constant $\lambda$ enforces a global and linear trade-off between correctness and brevity. Under GRPO, where parameter updates are driven by normalized advantage, this design may assign negative advantage to correct but longer trajectories. As a result, GRPO fails to distinguish such valid trajectories from incorrect ones, suppressing necessary exploratory reasoning and potentially degrading performance on complex tasks.

To address these issues, we aim to propose a new efficient reasoning reward that can dynamically adjust the calculation of the reward based on the relative difficulty of the question with respect to the model whose weights are being updated, while ensuring that correct trajectories are not incorrectly penalized.

## 3. *SmartThinker* Design

We propose *SmartThinker*, a GRPO-based efficient reasoning method with progressive CoT length calibration. Figure 2 shows an overview of our method. We first estimate the length and accuracy distribution through trajectories within the group, and calculate the optimal length with the highest accuracy. The estimated optimal length reflects the relative difficulty of the problem under the current policy. Then, we design a dynamic reward function to compress trajectory length by guiding correct but overly long trajectories toward the optimal length. Additionally, we introduce a dynamic length-reward coefficient to prevent the normalized advantages of correct trajectories from becoming negative, thereby avoiding confusion between overly long trajectories and incorrect ones.

### 3.1. Problem Formulation

Given a prompt $q$, we sample a group of $G$ reasoning trajectories $\{o_1, \ldots, o_G\}$ from the policy $\pi_\theta$. Each trajectory $o_i$ is a sequence of tokens with length $l_i$. Let $r_i^{\text{acc}} \in \{0, 1\}$ denote the correctness reward and $\boldsymbol{r}^{\text{acc}}$ represent the vector of all correctness rewards within a group. Let $\mathcal{L} = \{l_i \mid i \in [1, G]\}$, and $\mathcal{L}^{\text{acc}} = \{l_i \mid r_i^{\text{acc}} = 1, i \in [1, G]\}$ denotes the subset of the length of the correct trajectories within a group. Our objective is to optimize $\pi_\theta$ to maximize a composite reward $r$ that balances reasoning accuracy with efficiency, formulated as:

$$r_i = f\left(i, \boldsymbol{r}^{\text{acc}}, \mathcal{L}\right) \tag{4}$$

where $f$ accounts for the accuracy and length of both the current trajectory and other trajectories within the group.

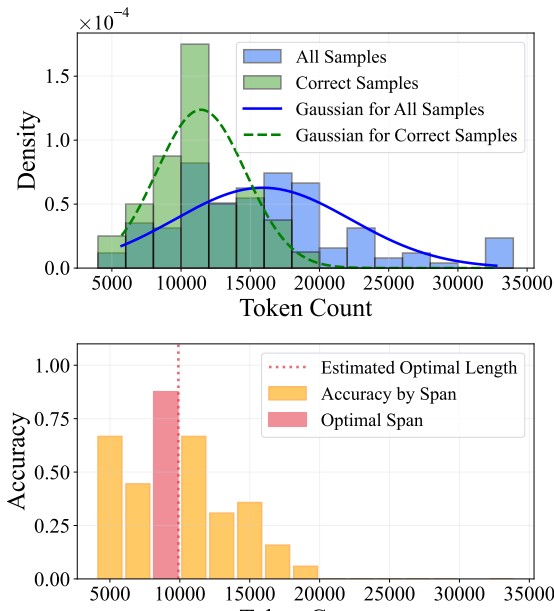

*Figure 3.* Estimation of the optimal length. The subfigure above shows the distribution of all samples and correct samples and the one below shows the span with the highest accuracy and the estimated optimal length.

### 3.2. Optimal Reasoning Length Derivation

Inspired by ShorterBetter (Yi et al., 2026), we calculate an optimal reasoning length for each prompt during rollout stages. ShorterBetter directly sets the optimal length to the shortest correct trajectory length. However, in fact, due to the randomness of model sampling, the shortest correct trajectory may lie in a very marginal position within the distribution of all correct trajectories, and directly approaching this length may lead to a decrease in model accuracy. Moreover, not all responses need to be compressed. When the responses to a question are already very short, or when the question is difficult while the model is under-reasoning, compressing the length may cause the model to lose its ability for accurate reasoning. We need to propose a new method for calculating the optimal length to achieve dynamic length compression with response distribution awareness.

Our optimal length calculation method is motivated by two observations. First, according to the previous study, LRMs exhibits peak accuracy in the responses to the same question across different lengths (Wu et al., 2025), so we propose that **the optimal length should be defined as the value that maximizes the conditional probability of correctness**, *i.e.*, $l^{\text{opt}} = \arg\max_l \Pr(r^{\text{acc}} = 1 \mid l, q; \theta)$. Secondly, we observe that the output length distribution of LRM for a given question, as well as the length of correct trajectories, often exhibits a distribution that is high in the middle and narrow at both ends. Figure 3 gives an example of the length distribution. Therefore, we consider modeling the

length distribution using a Gaussian distribution. Under the assumption that both the general length distribution and the successful length distribution follow Gaussian profiles, we derive the following theorem.

**Theorem 3.1.** *Let $\theta$ be the parameter of the current policy, $q$ be the input prompt. Assume that the distribution of $l$ under policy $\pi_\theta$ is $(l \mid q; \theta) \sim N(\mu_1, \sigma_1^2)$ and the distribution of $l$ given the correct reasoning condition is $(l \mid r^{acc} = 1, q; \theta) \sim N(\mu_2, \sigma_2^2)$. $\Pr\left(r^{acc} = 1 \mid l; \theta\right)$ has a unique finite maximum $\iff \sigma_1^2 > \sigma_2^2$, and the point is $\arg\max_l \Pr\left(r^{acc} = 1 \mid l; \theta\right) = \frac{\sigma_1^2 \mu_2 - \sigma_2^2 \mu_1}{\sigma_1^2 - \sigma_2^2}$.*

This theorem can be proved by obtaining the expression of $\Pr(r^{acc} = 1 \mid l, q; \theta)$ with respect to $l$ using the Bayesian formula, and then conducting monotonicity analysis through derivatives. The detailed proof of the theorem is given in Appendix A.1. We give an example of how the theorem works in Figure 3. It can be seen that the distributions of all samples and correct samples are both similar to a Gaussian distribution, and the estimated optimal length is very close to the optimal span with the highest accuracy. There are also two more extreme cases: $\sigma_1^2 < \sigma_2^2$ and $\sigma_1^2 = \sigma_2^2$. We provide the discussion of them in Appendix A.2. The complete formula for optimal length is:

$$
l^{\text{opt}} = \begin{cases} \frac{\sigma_1^2 \mu_2 - \sigma_2^2 \mu_1}{\sigma_1^2 - \sigma_2^2}, & \text{if } \sigma_1^2 > \sigma_2^2, \\ +\infty, & \text{if } \sigma_1^2 = \sigma_2^2 \text{ and } \mu_1 < \mu_2, \quad (5) \\ 0, & \text{otherwise.} \end{cases}
$$

### 3.3. Length Calibration with Distribution Estimation

Since GRPO samples $G$ trajectories for each prompt in each training round, we can estimate $\hat{\mu}_1$, $\hat{\mu}_2$, $\hat{\sigma}_1$, $\hat{\sigma}_2$ through the sampling results, *i.e.*, $\hat{\mu}_1 = \text{mean}(\mathcal{L})$, $\hat{\sigma}_1 = \text{std}(\mathcal{L})$, $\hat{\mu}_2 = \text{mean}(\mathcal{L}_i^{\text{acc}})$ and $\hat{\sigma}_2 = \text{std}(\mathcal{L}_i^{\text{acc}})$.

Based on the analysis above, we propose a new calculation method for optimal length:

$$
\hat{l}^{\text{opt}} = \text{clip}\left(l^*, \min\left(\mathcal{L}\right), \max\left(\mathcal{L}\right)\right), \quad (6)
$$

where

$$
l^* = \begin{cases} \frac{\hat{\sigma}_1^2 \hat{\mu}_2 - \hat{\sigma}_2^2 \hat{\mu}_1}{\hat{\sigma}_1^2 - \hat{\sigma}_2^2}, & \text{if } \hat{\sigma}_1^2 > \hat{\sigma}_2^2, \\ \max\left(\mathcal{L}\right), & \text{if } \hat{\sigma}_1^2 = \hat{\sigma}_2^2 \text{ and } \hat{\mu}_1 < \hat{\mu}_2, \quad (7) \\ \min\left(\mathcal{L}\right), & \text{otherwise.} \end{cases}
$$

It must be acknowledged that the Gaussian distribution assumption is highly idealized. However, from a heuristic perspective, $\hat{l}^{\text{opt}}$ adjusts reasoning length based on question difficulty relative to the policy. When the correct trajectory is generally longer, $\hat{l}^{\text{opt}} > \hat{\mu}$ corrects underthinking by encouraging depth; when the correct trajectory is generally shorter, $\hat{l}^{\text{opt}} < \hat{\mu}$ mitigates overthinking by promoting conciseness.

To make the model learn shorter reasoning paths, we only apply a length penalty to correct trajectories with length greater than the optimal length. For all incorrect trajectories, we do not apply a length reward. The length reward can be summarized as:

$$
r_i^{\text{len}} = \begin{cases} 0, & \text{if } r_i^{\text{acc}} = 0, \\ -\text{ReLU}\left(l_i - \hat{l}^{\text{opt}}\right), & \text{if } r_i^{\text{acc}} = 1. \end{cases} \quad (8)
$$

In the reward function, $\hat{l}^{\text{opt}}$ dynamically adjusts the proportion of compressed trajectories. When $\hat{l}^{\text{opt}} \geqslant \max\{\mathcal{L}^{\text{acc}}\}$, the length reward for all trajectories becomes 0, and *Smart-Thinker* degenerates into classical GRPO training. At this point, the training objective shifts from compressing length back to enhancing the model's reasoning capability.

### 3.4. Dynamic Length-Reward Coefficient

Since GRPO normalizes the reward to obtain the advantage, a static length-reward coefficient may result in negative advantages for overly long correct trajectories. To ensure that correct trajectories have non-negative advantages and incorrect trajectories have non-positive advantages, we design a dynamic length-reward coefficient.

The constraints can be defined as

$$
\begin{cases} 1 + \lambda r_i^{\text{len}} \geqslant \text{mean}(\boldsymbol{r}^{\text{acc}} + \lambda \boldsymbol{r}^{\text{len}}), & \text{if } r_i^{\text{acc}} = 1, \\ \lambda r_i^{\text{len}} \leqslant \text{mean}(\boldsymbol{r}^{\text{acc}} + \lambda \boldsymbol{r}^{\text{len}}), & \text{if } r_i^{\text{acc}} = 0, \end{cases} \quad (9)
$$

where $\boldsymbol{r}^{\text{len}}$ is the list of length rewards. Considering the length reward defined in Section 3.3, we have

$$
\begin{cases} r_i^{\text{len}} \leqslant 0, & \text{if } r_i^{\text{acc}} = 1, \\ r_i^{\text{len}} = 0, & \text{if } r_i^{\text{acc}} = 0. \end{cases} \quad (10)
$$

With Equation 10, when the first constraint of Equation 9 is satisfied, the second constraint is satisfied automatically. Solving the first constraint, we have

$$
0 < \lambda \leqslant \frac{p^{\text{err}}}{\text{mean}(\boldsymbol{r}^{\text{len}}) - \min\left(\boldsymbol{r}^{\text{len}}\right)}, \quad (11)
$$

where $p^{\text{err}}$ is the ratio of incorrect trajectories. To improve length compression efficiency, we define the dynamic length-reward coefficient as

$$
\begin{aligned} \Lambda\left(\boldsymbol{r}^{\text{acc}}, \boldsymbol{r}^{\text{len}}\right) &= \max \lambda \\ &= \frac{p^{\text{err}}}{\text{mean}(\boldsymbol{r}^{\text{len}}) - \min\left(\boldsymbol{r}^{\text{len}}\right)}. \end{aligned} \quad (12)
$$

### 3.5. Overall Reward and Advantage

With the coefficient calculated above, the total reward is calculated as follows:

$$
r_i = r_i^{\text{acc}} + \Lambda\left(\boldsymbol{r}^{\text{acc}}, \boldsymbol{r}^{\text{len}}\right) \cdot r_i^{\text{len}}. \quad (13)
$$

*Table 1.* Performance comparison across base models and benchmarks. The SmartThinker rows highlight our results.

| Method | #RL Steps ↓ | Math500 | | AIME25 | | AMC23 | | Average | | |
|---|---|---|---|---|---|---|---|---|---|---|
| | | Len. ↓ | Acc.(%) ↑ | Len. ↓ | Acc.(%) ↑ | Len. ↓ | Acc.(%) ↑ | Len. ↓ | Acc. (%) ↑ | AE ↑ |
| **DeepSeek-R1-Distill-Qwen-1.5B** | | | | | | | | | | |
| Base Model | N/A | 5420 | 84.9 | 15199 | 24.2 | 9320 | 73.1 | 9980 | 60.7 | N/A |
| Truncated Think 4k | N/A | 2950 | 71.4 | 4299 | 11.7 | 3627 | 42.5 | 3625 | 41.9 | -0.91 |
| Truncated Think 8k | N/A | 3971 | 72.0 | 7531 | 17.5 | 5471 | 50.6 | 5658 | 46.7 | -0.72 |
| ShorterBetter | 300 | 1008 | 71.0 | 3727 | 19.0 | 2246 | 66.9 | 2327 | 52.3 | 0.07 |
| ThinkPrune-4k | N/A | 2744 | 84.1 | 7462 | 22.5 | 4201 | **76.3** | 4802 | 60.95 | 0.53 |
| LASER-DE-4096 | 1000 | 2720 | **85.1** | 7706 | 22.5 | 4330 | 71.9 | 4919 | 59.8 | 0.42 |
| *SmartThinker* | **150** | 2645 | 84.5 | 8431 | **25.0** | 4421 | **76.3** | 5169 | **61.9** | **0.54** |
| **DeepSeek-R1-Distill-Qwen-7B** | | | | | | | | | | |
| Base Model | N/A | 3928 | 92.3 | 14829 | 35.0 | 6634 | 91.9 | 8463 | 73.1 | N/A |
| Truncated Think 4k | N/A | 2804 | 72.5 | 4434 | 10.8 | 3511 | 45.6 | 3583 | 43.0 | -1.48 |
| Truncated Think 8k | N/A | 3421 | 75.2 | 4303 | 13.3 | 4657 | 53.1 | 4127 | 47.2 | -1.25 |
| ShorterBetter | 200 | 1346 | 88.5 | 6409 | 30.0 | 2719 | 86.3 | 3491 | 68.3 | 0.26 |
| LASER-DE-4096 | 1000 | 1903 | **93.1** | 6578 | 30.0 | 2946 | **90.0** | 3809 | 71.0 | 0.41 |
| *SmartThinker* | **75** | 2753 | 93.0 | 9277 | **40.8** | 4118 | **90.0** | 5382 | **74.5** | **0.43** |
| **Qwen3-4B-Thinking-2507** | | | | | | | | | | |
| Base Model | N/A | 6680 | 97.0 | 21662 | 69.2 | 10777 | 99.4 | 13040 | 88.5 | N/A |
| Truncated Think 4k | N/A | 3287 | 44.3 | 4608 | 0.0 | 4319 | 12.0 | 4071 | 18.8 | -3.25 |
| Truncated Think 8k | N/A | 4829 | 43.0 | 4609 | 0.8 | 7067 | 15.0 | 5501 | 19.6 | -3.31 |
| *SmartThinker* | **50** | 3488 | **96.6** | 13761 | **71.7** | 5992 | **98.8** | 7747 | **89.0** | **0.42** |

With the reward function above, we can calculate trajectory-wise advantage as follows:

$$\hat{A}_i = \frac{r_i - \text{mean}\{r_j\}_{j=1}^{G}}{\text{std}\{r_j\}_{j=1}^{G}}. \tag{14}$$

The details of the GRPO algorithm we use is shown in Appendix A.3.

## 4. Experiment

### 4.1. Experiment Setup

**Datasets.** We post-train our base model on DeepScaleR-preview (Luo et al., 2025), a mathematical dataset comprising 40K problems of varying difficulty drawn from AIME, AMC, Omni-MATH (Gao et al., 2025a), and the STILL dataset (Nayab et al., 2024). No difficulty-based sampling is applied during training. We evaluate our method on three standard math benchmarks: MATH, AIME25, and AMC23.

**Models.** We evaluate our method on DeepSeek-R1-Distill-Qwen-1.5B, DeepSeek-R1-Distill-Qwen-7B (Guo et al., 2025), and Qwen3-4B-Thinking-2507 (Yang et al., 2025). The DeepSeek-R1-Distill models are widely adopted in prior efficient reasoning studies, and we therefore report direct comparisons with existing methods on these two models. In addition, we demonstrate the applicability and effectiveness of our approach to more recent architectures by fine-tuning Qwen3-4B-Thinking-2507.

**Training Configurations.** We implement *SmartThinker* using verl (Sheng et al., 2025). For all models, we use a batch size of 64, a group size of 8, a minibatch size of 16, and a maximum reasoning length of 8000 tokens. To improve training efficiency, we omit the KL loss. We fine-tune the 1.5B, 7B, and 4B models with a constant learning rate of $1 \times 10^{-6}$ for 150, 75, and 50 training steps, respectively.

**Evaluation.** All models and benchmarks were tested under the settings of temperature=0.6, top_p=1.0, max_tokens=32768. We sample 4 responses for each question in each benchmark. For each benchmark, we adopt three metrics: 1) **Accuracy (Acc.)** or Pass@1, which is defined as the ratio of correct responses among all responses; 2) **Length (Len.)**, the number of average output tokens; 3) **AE Score (AE)**, which is a balanced metric considering both accuracy and efficiency proposed by DeepScaleR (Luo et al., 2025). The detail description of the AE Score is provided in Appendix B.2.

**Baselines.** We compare against the base model, training-free truncation baselines, and representative single-stage RL methods optimized for length rewards. We adopt the following baselines: **Base Model**, **Truncated Think** $n$**k**, **ShorterBetter** (Yi et al., 2026), **ThinkPrune-4k** (Hou et al., 2025), **LASER-DE-4096** (Liu et al., 2025b). For all post-trained baselines, we directly use their open-sourced models. Except for ThinkPrune, all baseline models are trained on the DeepScaleR-Preview dataset (Luo et al., 2025), and the training set of ThinkPrune derives from historical AIME

*Table 2.* The result of combining AutoThink-Stage2 and *SmartThinker*.

| Method | #RL Steps ↓ | Math500 | | AIME25 | | AMC23 | | Average | | |
| --- | --- | --- | --- | --- | --- | --- | --- | --- | --- | --- |
| | | Len. ↓ | Acc.(%) ↑ | Len. ↓ | Acc.(%) ↑ | Len. ↓ | Acc.(%) ↑ | Len. ↓ | Acc. (%) ↑ | AE ↑ |
| Base Model | N/A | 5420 | 84.9 | 15199 | 24.2 | 9320 | 73.1 | 9980 | 60.7 | N/A |
| **AutoThink + *SmartThinker*** | | | | | | | | | | |
| AutoThink-Stage2 | 440 | 3593 | 85.4 | 10571 | 27.5 | 6206 | 75.0 | 6720 | 62.6 | 0.41 |
| AutoThink-Stage2 → Stage3 | 440 + 60 | 2196 | 85.2 | 9084 | 24.2 | 4117 | 73.8 | 5132 | 61.1 | 0.50 |
| **AutoThink-Stage2 → *SmartThinker*** | 440 + **50** | 2792 | 84.9 | 8760 | 29.2 | 5134 | 74.4 | 5562 | **62.8** | **0.55** |
| **ThinkPrune → *SmartThinker*** | | | | | | | | | | |
| ThinkPrune-4k | N/A | 2744 | 84.1 | 7462 | 22.5 | 4201 | 76.3 | 4802 | 61.0 | 0.53 |
| ThinkPrune-4k → 3k | N/A | 2224 | 84.0 | 5708 | 20.0 | 3104 | 75.0 | 3679 | 59.7 | 0.54 |
| ThinkPrune-4k → 3k → 2k | N/A | 1928 | 83.3 | 5010 | 16.7 | 2884 | 70.6 | 3274 | 56.9 | 0.35 |
| **ThinkPrune-4k → *SmartThinker*** | + 75 | 2471 | 85.1 | 6933 | 24.2 | 3939 | 74.4 | 4448 | **61.2** | **0.58** |

and AMC problems, which is also very similar, ensuring a relatively fair comparison in the experiment.

## 4.2. Main Results

We present our main results in Table 1. *SmartThinker* consistently achieves the strongest overall performance across all base models. On DeepSeek-R1 1.5B and 7B, it attains the highest average accuracy and AE score, demonstrating an optimal balance between accuracy and efficiency, while further improving model accuracy. Notably, it is the only method that improves average accuracy consistently across models of different scales.

**Difficulty-aware Length Adjustment.** *SmartThinker* exhibits adaptive inference lengths across benchmarks of varying difficulty. On relatively easier benchmarks such as MATH500, *SmartThinker* achieves high compression ratios, reducing unnecessary token waste. On more challenging benchmarks like AIME25, *SmartThinker* can achieve higher accuracy by generating more tokens. This suggests that *SmartThinker* enables models to produce problem-dependent reasoning lengths. And the model's efficient reasoning capability in 8k context training can be transferred to outputs exceeding 8k.

**Training Efficiency**. In addition to balancing accuracy and efficiency, *SmartThinker* also achieves training efficiency, reaching performance comparable or superior to other methods using only 150 and 75 training steps on 1.5B and 7B models respectively. Since our reward design does not update model weights based on completely correct or completely incorrect trajectories, our method actually has better training efficiency than what the training steps suggest. If difficulty-based sampling is performed before training, even fewer training steps could achieve the same effect. We also observe that for base models with stronger performance, *SmartThinker* requires fewer training steps, needing only 50 steps of training on Qwen3-4B-Thinking-2507. These results suggest that *SmartThinker* may achieve higher training efficiency as base models become stronger.

**Baseline Analysis.** While ShorterBetter achieves the shortest average reasoning length, it suffers substantial performance degradation: its reward fails to penalize overly short reasoning paths when answers are partially correct, allowing superficial reasoning to persist, and occasionally assigns positive advantage to fully incorrect trajectories, causing the model to learn from errors. In contrast, LASER-DE-4096 and ThinkPrune adopt less aggressive length reduction strategies, performing well on MATH500, but both experience notable accuracy drops on more challenging benchmarks such as AIME25.

## 4.3. Combination with Multi-Stage Frameworks

*SmartThinker* can not only be used as a standalone step to fine-tune LRM, but also integrated into other multi-stage efficient reasoning methods. We select AutoThink (Tu et al., 2026) and ThinkPrune (Hou et al., 2025). AutoThink is a typical multi-stage GRPO-based efficient reasoning method consisting of three stages. We replace the final stage of AutoThink with *SmartThinker* to train the 1.5B model, and compared it with the original method. ThinkPrune provides a series of models, including models trained in a single stage and models trained in multiple stages. ThinkPrune's paper shows that through multi-step training, the model's inference length can be further compressed, but the inference accuracy may decrease. We train the 1.5B model using *SmartThinker* based on ThinkPrune-4k and compare it with ThinkPrune-iter3k (4k → 3k) and iter2k (4k → 3k → 2k). The results are shown in Table 2. For both methods, *SmartThinker* outperforms the original models, demonstrating that our method can be flexibly inserted as a plug-in into the training phase of other approaches and can deliver better performance.

## 4.4. Training Process Analysis

We illustrate four different metrics during the training process in Figure 4. It can be observed that as training progresses, the model's accuracy gradually improves while the length continues to decrease. In addition, although we set

*Table 3.* OOD evaluation on models of different scales.

| Models | MMLU | | MathQA | | LiveCode | | HumanEval | | Average | |
|---|---|---|---|---|---|---|---|---|---|---|
| | Len. ↓ | Acc.(%) ↑ | Len. ↓ | Acc.(%) ↑ | Len.↓ | Acc.(%) ↑ | Len. ↓ | Acc.(%) ↑ | Len. ↓ | Acc.(%) ↑ |
| **DeepSeek-R1-Distill-Qwen-1.5B** | | | | | | | | | | |
| Base Model | 1686 | 46.02 | 5206 | 84 | 10668 | 24.2 | 4740 | 68.9 | 5575 | 55.78 |
| *SmartThinker* | 994 | 46.48 | 2581 | 84.6 | 7658 | 25.4 | 3099 | 69.5 | 3583 | 56.495 |
| **Qwen3-4B-Thinking-2507** | | | | | | | | | | |
| Base Model | 2978 | 80.8 | 3253 | 91.3 | 12388 | 68.1 | 4507 | 95.7 | 5781.5 | 83.975 |
| *SmartThinker* | 2167 | 81.1 | 2224 | 91.7 | 9376 | 68.4 | 3158 | 96.5 | 4231.25 | 84.425 |

── 1.5B model  ── 7B model  ── 4B model

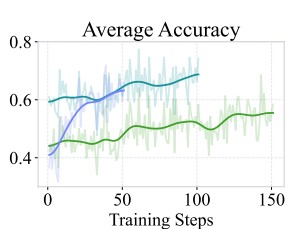 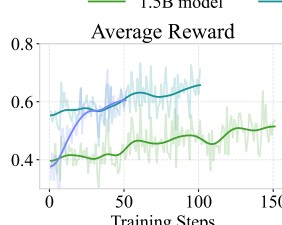 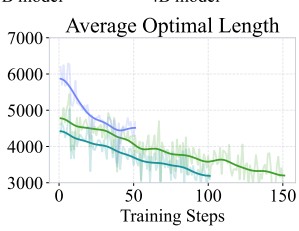 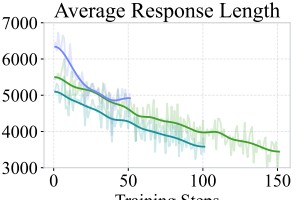

*Figure 4.* Changes in various metrics during the training process

a dynamically changing reward function, the reward can still steadily increase during training, consistent with the expectations of reinforcement learning.

We also observe an interesting phenomenon: during training, the optimal length is generally lower than the output length. This implies: 1) overthinking in model responses is widespread; 2) as the policy updates, the model's optimal length dynamically changes, supporting the necessity of setting dynamic length rewards; 3) simply making the output length approach the optimal length can effectively shorten the CoT.

### 4.5. Out-of-Domain Test

To assess whether the efficient reasoning behavior learned from mathematical reasoning tasks generalizes to other domains without compromising model accuracy, we also conduct out-of-domain (OOD) tests on three additional benchmarks: MathQA (Amini et al., 2019), MMLU (Hendrycks et al., 2020b;a), LiveCodeBench (Jain et al., 2025), and HumanEval (Chen, 2021).

We test OOD benchmarks on the 1.5B and 4B models. The comparison between *SmartThinker* and the base models of both 1.5B and 4B is shown in Table 3. The results show that both models can achieve accuracy improvements while compressing sequence length on all OOD benchmarks, demonstrating the generalization capability of *SmartThinker*. This also indicates that the accuracy improvement brought by *SmartThinker* does not stem from training on more math problems, but rather from enabling the model to learn beneficial reasoning paths through reasonable reward allocation.

We also provide a comparison between *SmartThinker* and

*Table 4.* Ablation study on DeepSeek-R1-Distill-Qwen-1.5B

| Method | Math500 | | AIME25 | | AMC23 | | Average | |
|---|---|---|---|---|---|---|---|---|
| | len | acc | len | acc | len | acc | len | acc |
| Fixed Coefficient | 1897 | 80.3 | 6049 | 21.7 | 2987 | 70.6 | 3644 | 57.5 |
| Symmetric | 2902 | 83.4 | 8761 | 24.2 | 4927 | 72.9 | 5530 | 60.2 |
| Linear | 2439 | 83 | 6909 | 21.7 | 3379 | 70 | 4242 | 58.2 |
| *SmartThinker* | 2645 | **84.5** | 8431 | **25** | 4421 | **76.3** | 5169 | **61.9** |

all baselines. The results are shown in Appendix C.3.

### 4.6. Ablation Study

To evaluate the contributions of the dynamic length reward and dynamic length-reward coefficient, we consider three reward configurations for the ablation study: **Fixed Coefficient**, **Symmetric**, **Linear**. The details of these configurations are shown in Appendix B.4.

We present the results in Table 4. The results show that *SmartThinker* outperforms all ablation settings, indicating that each component contributes to the final performance. It is worth noting that under the **Symmetric**, although the output length is reduced compared to the base model, there remains a gap in both efficiency and accuracy compared to *SmartThinker*, indicating that while forcing all correct trajectories to approach the optimal length can reduce length, simply shortening excessively long trajectories yields better results.

### 4.7. *SmartThinker*'s Impact on CoT Structure

To quantify the optimization effect of the model's chain-of-thought structure, we have additionally used LLM-as-a-judge to classify the model's reasoning structure. We use DeepSeek-V3.2 (Liu et al., 2025a) as the judge model and employ the same system prompt as in ShorterBetter (Yi et al.,

*Table 5. SmartThinker's impact on CoT structure.*

| Category | Base Model | *SmartThinker* |
|---|---|---|
| Pivotal Reasoning | 6.13% | 16.04% |
| Repetition & Rephrasing | 1.13% | 0.68% |
| Exploring Alternatives | 86.86% | 77.29% |
| Verification & Explanation | 2.73% | 2.90% |
| Non-Substantive Statement | 3.05% | 3.09% |

2026). We examine the reasoning structures of the 1.5B models on 50 randomly selected problems from math500 before and after training, which is presented in Table 5. The results show that after training, the model produces fewer exploratory detours and repetitive statements, while increasing the proportion of key reasoning paths. This result indicates that even without explicit process rewards, simply allocating sparse outcome rewards appropriately can improve the structure of CoT.

# 5. Related Works

**Large Reasoning Models (LRMs).** The academic community has discovered that designing prompts to guide models to output chain-of-thought(CoT) can improve the accuracy of non-reasoning models (Wei et al., 2022). The emergence of OpenAI o1 (Jaech et al., 2024) illustrates LRMs' capabilities on complicated logical tasks like mathematics, coding, and scientific problems. As a groundbreaking advancement, DeepSeek-R1 (Luo et al., 2025) provides an efficient and scalable training method for LRM, sparking a surge in reinforcement learning-based LRM training. Nowadays, whether open-source large models (Yang et al., 2025; Liu et al., 2025a; Zeng et al., 2025) or closed-source commercial models (Comanici et al., 2025), LRM has become a mature new paradigm for large language models.

**RL-Based Post Training of LLMs.** Reinforcement Learning (RL) has become the standard for aligning LLMs with complex reasoning tasks. The main algorithms of LLM in RL include Proximal Policy Optimization(PPO) (Schulman et al., 2017; Ouyang et al., 2022), Direct Preference Optimization(DPO) (Rafailov et al., 2023), etc. The introduction of Group Relative Policy Optimization (GRPO) (Shao et al., 2024) provides a simpler and more effective method for LRM training by removing the need for a separate value model and instead using group-wise relative rewards. There have already been many improved algorithms based on GRPO, such as DAPO (Yu et al., 2026), GSPO (Zheng et al., 2025), SAPO (Gao et al., 2025b), and so on.

**Efficient LLM Reasoning.** There have been many methods focusing on efficient LLM reasoning, including training-free and training-based methods. Training-free methods involve techniques such as truncating the chain-of-thought, selecting the optimal path by single-step sampling (Xu et al.,

2026a), and using spline vectors to induce fast thinking (Lin et al., 2026). These methods usually have limited compression or introduce additional overhead. Training-based methods (Yi et al., 2026; Liu et al., 2025b; Hou et al., 2025) can achieve more flexible length control by designing training objectives, while even improving accuracy.

# 6. Conclusion and Limitations

In this paper, we propose *SmartThinker*, a novel GRPO-based efficient reasoning method that shortens the CoT length of LRMs by progressive CoT length calibration. We derive an estimate of the reasoning length that maximizes the probability of correct answers through probabilistic modeling. During training, we estimate the optimal length based on the distribution of response lengths and correctness for each question, and implement dynamic length calibration by guiding overly long answers to approach the optimal length. We further design a dynamic length-reward coefficient to prevent correct trajectories from receiving negative advantages. We validate the effectiveness of *SmartThinker* through extensive experiments, demonstrating that *SmartThinker* can improve the model's reasoning capability while compressing the CoT length, achieving a balance between efficiency and accuracy.

Although we have verified the effectiveness of *SmartThinker* on GRPO, its performance on other variants of GRPO remains to be validated, and the effectiveness of *SmartThinker* on some open-ended questions is still difficult to assess. Moreover, like other methods, *SmartThinker* is still limited to designing outcome-based rewards, lacking process reward supervision, making it difficult to discover fine-grained beneficial reasoning patterns during the reasoning process. Combining *SmartThinker* with fine-grained process rewards is a promising direction for future work.

## Acknowledgements

This work was sponsored in part by China NSF grant No. U25A6024, 62472278, 62432007, 62441236, 62332014, 62332013, and 62225202, and Shanghai QiYuan Innovation Foundation. This work is partially supported by SJTU Kunpeng & Ascend Center of Excellence. The opinions, findings, conclusions, and recommendations in this paper are those of the authors and do not necessarily reflect the views of the funding agencies or the government.

## Impact Statement

This paper presents work whose goal is to advance the field of machine learning. There are many potential societal consequences of our work, none of which we feel must be specifically highlighted here.

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

# A. Math Supplement

## A.1. Proof of Theorem 3.1

In this section, we provide the detailed proof process of Theorem 3.1

> **Theorem 3.1**
>
> Let $\theta$ be the parameter of the current policy, $q$ be the input prompt. Assume that the distribution of $l$ under policy $\pi_\theta$ is $(l \mid q; \theta) \sim N(\mu_1, \sigma_1^2)$ and the distribution of $l$ given the correct reasoning condition is $(l \mid r^{acc} = 1, q; \theta) \sim N(\mu_2, \sigma_2^2)$. $\Pr(r^{acc} = 1 \mid l; \theta)$ has a unique finite maximum $\iff \sigma_1^2 > \sigma_2^2$, and the point is $\arg\max_l \Pr(r^{acc} = 1 \mid l; \theta) = \frac{\sigma_1^2 \mu_2 - \sigma_2^2 \mu_1}{\sigma_1^2 - \sigma_2^2}$.
>
> ------
>
> *Proof.* By Bayes' theorem, the posterior probability of correct reasoning given the length $l$ is expressed as:
>
> $$\Pr(r^{acc} = 1 \mid l, q; \theta) = \frac{p(l \mid r^{acc} = 1, q; \theta) \Pr(r^{acc} = 1 \mid q; \theta)}{p(l \mid q; \theta)} \tag{15}$$
>
> Since $\Pr(r^{acc} = 1 \mid \theta)$ is independent of $l$, the optimization objective satisfies:
>
> $$\arg\max_l \Pr(r^{acc} = 1 \mid l, q; \theta) = \arg\max_l \frac{p(l \mid r^{acc} = 1, q; \theta)}{p(l \mid q; \theta)} \tag{16}$$
>
> Substituting the probability density functions for $N(\mu_2, \sigma_2^2)$ and $N(\mu_1, \sigma_1^2)$, we obtain:
>
> $$\frac{p(l \mid r^{acc} = 1, q; \theta)}{p(l \mid q; \theta)} = \frac{\sigma_1}{\sigma_2} \exp\left(\frac{(l - \mu_1)^2}{2\sigma_1^2} - \frac{(l - \mu_2)^2}{2\sigma_2^2}\right) \tag{17}$$
>
> Taking the natural logarithm and omitting terms constant with respect to $l$, we define the objective function:
>
> $$f(l) = \frac{(l - \mu_1)^2}{2\sigma_1^2} - \frac{(l - \mu_2)^2}{2\sigma_2^2} \tag{18}$$
>
> The first-order condition for the maximum is $\frac{df}{dl} = 0$:
>
> $$\frac{l - \mu_1}{\sigma_1^2} - \frac{l - \mu_2}{\sigma_2^2} = 0 \implies (l - \mu_1)\sigma_2^2 = (l - \mu_2)\sigma_1^2 \tag{19}$$
>
> Solving for $l$ yields the critical point:
>
> $$l^{\text{opt}} = \frac{\sigma_1^2 \mu_2 - \sigma_2^2 \mu_1}{\sigma_1^2 - \sigma_2^2} \tag{20}$$
>
> To confirm this is a maximum, we examine the second derivative:
>
> $$\frac{d^2 f}{dl^2} = \frac{1}{\sigma_1^2} - \frac{1}{\sigma_2^2} \tag{21}$$
>
> If and only if $\sigma_1^2 > \sigma_2^2$, it follows that $\frac{d^2 f}{dl^2} < 0$, which ensures that $l^{\text{opt}}$ is a global maximum. $\square$

## A.2. In-depth Analysis of the Optimal Length $l^{\text{opt}}$

To better understand the properties of the proposed optimal reasoning length $l^{\text{opt}}$, we analyze the log-conditional-probability $g(l) = \ln \Pr(r^{\text{acc}} = 1 \mid l)$. According to Bayes' theorem, $\Pr(r^{\text{acc}} = 1 \mid l) = \frac{\Pr(l \mid r^{\text{acc}} = 1) \Pr(r^{\text{acc}} = 1)}{\Pr(l)}$. Maximizing this probability with respect to $l$ is equivalent to maximizing the log-density ratio $h(l) = \ln \frac{p_2(l)}{p_1(l)}$, where $p_1, p_2$ are the probability density functions of $N(\mu_1, \sigma_1^2)$ and $N(\mu_2, \sigma_2^2)$ respectively. Figure 5 simply illustrates how the curve of $h(l)$ changes under different conditions.

*Figure 5.* The curve of $h(l)$ under different conditions.

**Curvature and Stability.** The second derivative of the objective function is:

$$\frac{\partial^2}{\partial l^2} \ln \frac{p_2(l)}{p_1(l)} = \frac{1}{\sigma_1^2} - \frac{1}{\sigma_2^2}. \tag{22}$$

- **Concentration Case** ($\sigma_1^2 > \sigma_2^2$): The second derivative is negative, indicating that $l^* = \frac{\sigma_1^2 \mu_2 - \sigma_2^2 \mu_1}{\sigma_1^2 - \sigma_2^2}$ is a unique *maximum point*. In LLM reasoning, this is the most common scenario: while incorrect trajectories vary significantly (lengthy loops or abrupt failures), correct trajectories tend to concentrate within a specific logical complexity. The policy is thus encouraged to stay within this "reliable" length zone.

- **Dispersion Case** ($\sigma_1^2 < \sigma_2^2$): $l^* = \frac{\sigma_1^2 \mu_2 - \sigma_2^2 \mu_1}{\sigma_1^2 - \sigma_2^2}$ becomes a *minimum point*. This implies that the correctness probability is higher at extreme lengths (very short or very long) and lowest at $l^*$. For variable stability, we clip these cases to the observed boundaries in our implementation.

- **Degenerate Case** ($\sigma_1^2 = \sigma_2^2$): Due to the randomness of parameter estimation, this is an almost impossible scenario. However, for the sake of rigor, we discuss this scenario in detail below.

**Analysis of the Degenerate Case** ($\sigma_1^2 = \sigma_2^2$). Although the exact equality $\sigma_1 = \sigma_2$ is unlikely in practice with sampled data, it represents a critical theoretical threshold where the curvature of the log-probability ratio vanishes. When variances are identical, the quadratic terms in the exponent of the Gaussian distributions cancel out:

$$h(l) = \ln \frac{p_2(l)}{p_1(l)} = \frac{1}{2\sigma^2} \left[ (l - \mu_1)^2 - (l - \mu_2)^2 \right] + C = \frac{\mu_2 - \mu_1}{\sigma^2} l + C', \tag{23}$$

where $C$ and $C'$ are constants independent of $l$. In this case, the relationship between length and correctness becomes *strictly monotonic*:

- If $\mu_1 > \mu_2$, the slope is negative, implying that every additional token strictly decreases the likelihood of success. This suggests that the model should adopt a "minimalist" strategy ($l^{\text{opt}} \to 0$).

- If $\mu_1 < \mu_2$, the slope is positive, suggesting that accuracy is directly proportional to reasoning length, pushing the optimal target to be as long as possible ($l^{\text{opt}} \to \infty$).

- If $\mu_1 = \mu_2$ and $\sigma_1^2 = \sigma_2^2$, then $h(l)$ is constant, meaning length provides no information about correctness.

This linear degradation highlights the importance of the variance term in our theorem. When $\sigma_1^2 > \sigma_2^2$, the variance difference provides the "repelling force" that creates a stable, finite peak for the optimal length, preventing the model from collapsing into trivial (zero length) or infinitely redundant outputs.

**Geometric Interpretation of the Shift.** The optimal length can be rewritten as a biased estimate of the mean of correct trajectories:

$$l^{\text{opt}} = \mu_2 + \underbrace{\frac{\sigma_2^2}{\sigma_1^2 - \sigma_2^2} (\mu_2 - \mu_1)}_{\text{Correction Bias}}. \tag{24}$$

This decomposition reveals how the disparity between correct and total samples shifts the target:

- If $\mu_2 < \mu_1$ (correct samples are shorter on average), the correction bias is negative. Thus, $l^{\text{opt}}$ will be *smaller* than the mean correct length $\mu_2$. This mathematically justifies a "strict efficiency" penalty, where the model is pushed to be even more concise than the current average success.

- The term $\frac{\sigma_2^2}{\sigma_1^2 - \sigma_2^2}$ acts as a "variance leverage". If the correct trajectories are highly consistent ($\sigma_2^2 \to 0$), then $l^{\text{opt}} \approx \mu_2$, relying directly on the empirical success mean.

**Summary of Parameter Relationships.** Table 6 summarizes the theoretical behavior of $l^*$ under different statistical conditions.

*Table 6.* Behavior of $l^{\text{opt}}$ under different $\mu$ and $\sigma$ conditions.

| Condition | Mathematical Property | Reasoning Intuition | $l^{\text{opt}}$ Strategy |
|---|---|---|---|
| $\sigma_1^2 > \sigma_2^2$ | Local Maximum | Existence of a "Golden Length" | Computed $l^{\text{opt}}$ |
| $\sigma_1^2 = \sigma_2^2, \mu_1 > \mu_2$ | Linear Decrease | Shorter is strictly better | 0 (min length) |
| $\sigma_1^2 = \sigma_2^2, \mu_1 < \mu_2$ | Linear Increase | Complexity aids accuracy | $+\infty$ (max length) |
| $\sigma_1^2 < \sigma_2^2$ | Local Minimum | Unstable middle-ground | 0 (boundary clipping) |

### A.3. Details of Token-Wise GRPO Algorithm

Here we provide a complete version of GRPO objective with token-mean:

$$\mathcal{J}_{\text{GRPO}}(\theta) = \mathbb{E}_{q \sim P(Q), \{o_i\}_{i=1}^{G} \sim \pi_{\text{old}}(\cdot | q)} \left\{ \frac{1}{\sum_{i=1}^{G} |o_i|} \sum_{i=1}^{G} \sum_{t=1}^{|o_i|} \right.$$

$$\left. \left[ \min \left( r_{i,t}(\theta) \hat{A}_{i,t}, \text{clip} \left( r_{i,t}(\theta), 1-\epsilon, 1+\epsilon \right) \hat{A}_{i,t} \right) - \beta \mathbb{D}_{\text{KL}} \left( \pi_\theta \left( o_{i,t} \right) || \pi_{\text{ref}} \left( o_{i,t} \right) \right) \right] \right\}, \tag{25}$$

where

$$r_{i,t}(\theta) = \frac{\pi_\theta \left( o_{i,t} | q, o_{i,<t} \right)}{\pi_{\text{old}} \left( o_{i,t} | q, o_{i,<t} \right)}, \tag{26}$$

$$\mathbb{D}_{\text{KL}} \left( \pi_\theta \left( o_{i,t} \right) || \pi_{\text{ref}} \left( o_{i,t} \right) \right) = \frac{\pi_{\text{ref}} \left( o_{i,t} | q, o_{i,<t} \right)}{\pi_\theta \left( o_{i,t} | q, o_{i,<t} \right)} - \log \frac{\pi_{\text{ref}} \left( o_{i,t} | q, o_{i,<t} \right)}{\pi_\theta \left( o_{i,t} | q, o_{i,<t} \right)} - 1. \tag{27}$$

Considering training efficiency, $\beta$ can be set to 0 to reduce the overhead of computing $\pi_{\text{ref}}$.

## B. Evaluation Details

### B.1. Prompt

For both training and evaluation, we use the following prompt:

> **Prompt**
>
> {question}
> Please reason step by step, and put your final answer within \boxed{}.

### B.2. AE Score

The AE score is defined as

$$\text{AE} = \begin{cases} -\phi \cdot \Delta\text{Len.} + \eta \cdot \Delta\text{Acc.} & \text{if } \Delta\text{Acc.} \geqslant 0, \\ -\phi \cdot \Delta\text{Len.} + \theta \cdot \Delta\text{Acc.}, & \text{if } \Delta\text{Acc.} < 0, \end{cases} \tag{28}$$

*Table 7.* Gaussian test of length distributions.

|  | $P_{\text{SW}}$ | $P_{\text{KS}}$ and $P_{\text{shape}}$ | $P_{\text{KS}}$ |
|---|---|---|---|
| All Responses | 20.0 | 52.3 | 62.9 |
| Correct Responses | 27.7 | 60.0 | 74.3 |

where $\phi, \eta, \theta$ are all positive hyperparameters. To penalize performance degradation, generally $\theta > \eta$. We follow the default setting in DeepScaleR, *i.e.*, $\phi = 1, \eta = 3, \theta = 5$.

### B.3. Baselines

- **Base Model:** We evaluate the performance of the original models without RL finetuning.

- **Truncated Think $n$k:** This means setting a maximum thinking length for the base model's CoT. When the chain of thought reaches the maximum length, directly output the `</think>` token to end thinking and produce the final answer. We test this method to evaluate the performance of simple efficient reasoning approaches without training.

- **ThinkPrune-4k:** ThinkPrune is a GRPO-based efficient reasoning method which adopts length-pruning strategy in the reward design. We choose the model with the highest accuracy in ThinkPrune series.

- **ShorterBetter:** Differing from the naive length reward where trajectories within a group are independent, ShorterBetter designs two different non-monotonic reward functions for the cases of all answers being wrong and at least one correct answer existing. This approach can greatly compress the chain-of-thought length.

- **LASER-DE-4096:** This method comes from the LASER series. LASER designs a difficulty-aware reward, assigning different reward functions to different difficulty levels. We select LASER-DE-4096 as the baseline in this series of models because it strikes a good balance between performance and efficiency.

### B.4. Ablation Study Settings

- **Fixed Coefficient.** Do not use dynamic coefficient, directly setting $\lambda = 0.001$. The overall reward is $r_i = r_i^{\text{acc}} + \lambda r_i^{\text{len}}$

- **Symmetric.** We have analyze that directly pushing all output length approaching optimal length can also reduce the length, so here we directly set the length reward as:

$$r_i^{\text{len}} = \begin{cases} 0, & \text{if } r_i^{\text{acc}} = 0, \\ -|l_i - l^{\text{opt}}|, & \text{if } r_i^{\text{acc}} = 1, \end{cases} \tag{29}$$

- **Linear.** A simple length reward setting, which is

$$r_i^{\text{len}} = \begin{cases} 0, & \text{if } r_i^{\text{acc}} = 0, \\ l_i - \min(\mathcal{L}) & \text{if } r_i^{\text{acc}} = 1, \end{cases} \tag{30}$$

## C. Additional Experiments

### C.1. Gaussian Distribution Test of All Response Lengths and Correct Response Lengths

To verify the reasonableness of our theoretical analysis assumptions, we conducted a Gaussian distribution test on the model's response length. We use **Shapiro-Wilk** and **Kolmogorov-Smirnov** methods to test the distributions. For both methods, the significance threshold was set to 0.05. In addition, we also use skewness and kurtosis to detect shape. We denote passing the Shapiro-Wilk test as $P_{\text{SW}}$, passing the Kolmogorov-Smirnov test as $P_{\text{KS}}$, and satisfying absolute skewness less than 1.5 and absolute kurtosis less than 3.0 as $P_{\text{shape}}$.

We generate 64 responses for each question on the Olympiad (He et al., 2024) dataset and filter out questions with fewer than 16 correct answers. The length distribution test of the remaining questions is shown in the Table 7. The results show that, under tolerable shape deviations, more than half of the responses to the questions can be approximated by a Gaussian distribution. This proves the reasonableness of the premises in our theoretical analysis.

*Table 8.* Unimodal test results

|  | Model | Pass Rate |
| --- | --- | --- |
| All Responses | DeepSeek-R1-Distill-1.5B | 89.90% |
|  | SmarThinker-Distill-1.5B | 94.50% |
| Correct Responses | DeepSeek-R1-Distill-1.5B | 98.00% |
|  | SmarThinker-Distill-1.5B | 100% |

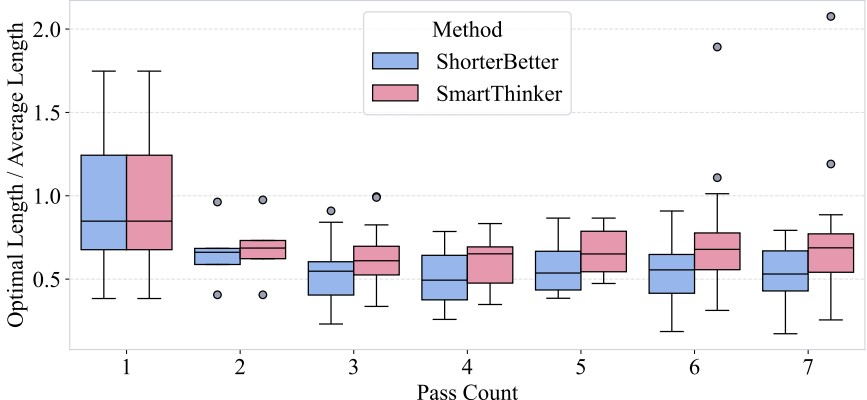

*Figure 6.* Comparison of optimal length under different pass counts.

We also performed Hartigan's dip test on samples that failed the Kolmogorov-Smirnov test to assess their unimodality. The results are shown in Table 8, which indicates that the vast majority of length distributions do not exhibit multimodal properties. Length distributions that failed the Gaussian test usually did so because of a relatively uniform distribution or peak shift. In such cases, using the mean and variance to describe these distributions remains effective.

### C.2. Optimal Length Comparison between ShorterBetter and *SmartThinker*

We compare the optimal length of ShorterBetter and *SmartThinker* under different pass counts using DeepSeek-R1-Distill-Qwen-1.5B, which is presented in Figure 6. Since the model's reasoning length is generally proportional to the question difficulty, for ease of comparison, we show the ratio of optimal length to average length in the figure. As can be seen from the figure, the optimal length of *SmartThinker* is significantly higher than that of ShorterBetter. This indicates that the length compression strategy of *SmartThinker* is more moderate. From the benchmark tests in Table 1, it can be observed that this more moderate compression strategy can maintain the model's high-quality reasoning ability while achieving compression.

We also observe that the optimal length does not change monotonically with the pass rate, indicating that the causes of model errors are diverse, possibly due to either overthinking or underthinking. This further highlights the importance of dynamically computing the optimal length.

### C.3. Out-of-Domain Tests on All Methods

We test the 1.5B models of all methods on MathQA (Amini et al., 2019), MMLU (Hendrycks et al., 2020b;a), Live-CodeBench (Jain et al., 2025), and HumanEval (Chen, 2021). The results are shown in Figure 7, which shows that all methods except *SmartThinker* maintain or even improve accuracy while compressing output length, achieving comparable levels of efficiency and accuracy. We also counted the number of MMLU subjects in which each method ranked first, which is presented in Figure 8. The figure shows that *SmartThinker* achieved the most first-place rankings, accounting for as high as 42%. This indicates that the capability learned by LRMs on math problems can be transferred to other domains.

### C.4. Length Distribution on Math500 of Varying Difficulty

We test the reasoning length compression ratio of questions with different difficulty levels using the built-in difficulty labels of Math500 on a 1.5B model. The results are shown in Figure 9 It can be observed that under the 1.5B parameter scale, our method achieves the highest compression rate across all difficulty levels. We also notice that the compression ratio

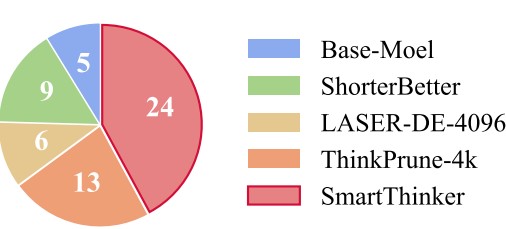

*Figure 7.* Test results of baseline and *SmartThinker* on four out-of-domain benchmarks.

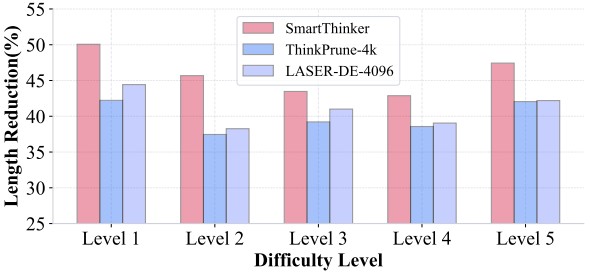

*Figure 8.* #Rank1 MMLU subjects of each method.

*Figure 9.* Compression ratios of different methods on problems of various difficulty levels in Math500.

of *SmartThinker* decreases as difficulty increases from level 1 to 4, but shows a significant rise at level 5, while the other methods exhibit more random length distributions. This indicates that models trained with the *SmartThinker* approach tend to increase output length for harder questions to improve accuracy, whereas for extremely difficult questions, due to a significant drop in correctness, the model shortens its output to avoid unnecessary token waste.

# D. Demonstration of an Example

In this section, we use a case to demonstrate the differences between the model trained with *SmartThinker* and the base model in terms of reasoning length and thinking structure. The example is shown in the boxes below.

In the example, the sequence generated by the base model uses fewer tokens in the Plan stage but repeatedly attempts during exploration, resulting in significant token consumption. In contrast, the sequence generated by *SmartThinker* consumes more tokens in the Plan stage, enabling the model to derive the correct answer with fewer tokens during the exploration phase. This indicates that models trained with *SmartThinker* optimize the structure of their chain-of-thought, performing thorough planning before attempting solutions, thereby streamlining the reasoning process while improving accuracy.

---

### Question

The 9 members of a baseball team went to an ice cream parlor after their game. Each player had a singlescoop cone of chocolate, vanilla, or strawberry ice cream. At least one player chose each flavor, and the number of players who chose chocolate was greater than the number of players who chose vanilla, which was greater than the number of players who chose strawberry. Let $N$ be the number of different assignments of flavors to players that meet these conditions. Find the remainder when $N$ is divided by 1000.

- - -

Ground Truth: 16

---

**Base Model: 16404 Tokens**

**Understand the Question: 94 Tokens**
<think>
Okay, so I have this problem where 9 baseball team members each ordered a singlescoop ice cream $\cdots$ I need to find the number of different assignments that meet these conditions and then find the remainder when that number is divided by 1000.

- - -

**Plan: 134 Tokens**
Hmm, okay. Let's break this down step by step. First, without any restrictions $\cdots$
So, step one is to compute the number of $\cdots$

- - -

**Initial Attempt: 5375 Tokens**
So, the total number of onto functions is given by the inclusion-exclusion principle $\cdots$
Wait, hold on, c=4, v + s=5 $\cdots$
Wait, hold on. Wait, we have to make sure $\cdots$
Wait, hold on, if c=3, then v must be 2 or 1 $\cdots$
But that might be more complicated.

- - -

**Repeated Attempts: 10025 Tokens**
Alternatively, perhaps using generating functions or other combinatorial techniques.
Wait, another approach: $\cdots$
Alternatively, perhaps the number of $\cdots$
Alternatively, perhaps I can compute $\cdots$
Therefore, the answer is 422.

- - -

**Double-Check: 311 Tokens**
But, in the 5-player case, 50 is the number of assignments where c >v >s$\cdots$
Therefore, the answer is 422.
But I'm not entirely confident$\cdots$
Therefore, I think the answer is 422.
</think>

- - -

**Answer: 568 Tokens**
To solve the problem, we need to find $\cdots$
Thus, the final answer is $\boxed{422}$.

---

*SmartThinker*: 2935 Tokens

**Understand the Problem: 108 Tokens**
<think>
Alright, so I have this problem where 9 players from a baseball team went to an ice cream parlor $\cdots$
I need to find the number of different assignments of flavors to players that meet these conditions, and then find the remainder when that number is divided by 1000.

- - -

**Plan: 487 Tokens**
Okay, let me try to break this down step by step.
First, without any restrictions $\cdots$
But wait, it's not just about the counts $\cdots$
So, the problem reduces to:
$\cdots$

- - -

**Explore the Solution: 1564 Tokens**
So, first step is to find all ordered triples (C, V, S) with C >V >S $\geq$ 1 and C + V + S = 9.
$\cdots$
Wait, but since C >V >S, each triple is ordered $\cdots$
Wait, no, actually, for each such partition $\cdots$ Wait, no, a + b + c = 9, so if a is as large as possible $\cdots$
$\cdots$
Thus, the remainder is 16.

- - -

**Double-Check: 269 Tokens**
Wait, is that it? That seems straightforward.
But let me double-check the calculations. $\cdots$
Therefore, the answer is 16.
**Final Answer**
$\boxed{16}$
</think>

- - -

**Answer: 507 Tokens**
The problem involves finding the number of $\cdots$
Thus, the remainder when $N$ is divided by 1000 is $\boxed{16}$.

