# OpenReview forum: "SmartThinker: Progressive Chain-of-Thought Length Calibration for Efficient Large Language Model Reasoning"
_ICML.cc/2026/Conference — ICML 2026 regular_

### Official Review · Reviewer_wBsb · 2026-03-06

**Soundness:** 3
**Presentation:** 3
**Significance:** 3
**Originality:** 2
**Overall Recommendation:** 4
**Confidence:** 3

**Summary:**

The paper addresses the efficiency and overthinking problems observed in Large Reasoning Models like OpenAI o1 and DeepSeek-R1. While long CoT processes enhance performance, they often result in redundancy and excessive computational costs. The authors proposing a method called SmartThinker. Unlike existing GRPO methods that apply static, linear length penalties often leading to over-compression and accuracy loss, SmartThinker introduces a dynamic length calibration mechanism.

**Compliance With Llm Reviewing Policy:**

Affirmed.

**Key Questions For Authors:**

Please see "Weaknesses".

**Limitations:**

Yes

**Strengths And Weaknesses:**

Strengths:

1. The paper moves beyond the heuristic "shorter is better" assumption. By modeling the conditional probability of correctness given length using Gaussian distributions, the authors provide a principled justification for their optimization target. This probabilistic approach to finding is mathematically grounded and intuitively appealing.

2. The authors present a static length rewards fail to account for problem difficulty. SmartThinker addresses this by dynamically adjusting the target length. As shown in the experiments, the model learns to compress easy problems significantly while retaining length for harder problems, avoiding the over-compression trap of methods like Truncated Think or static penalties.

3. The results are impressive. Achieving a ~50% token reduction is valuable, but doing so while improving accuracy by such a large margin on challenging benchmarks is significant. The comparison against competitive baselines is comprehensive.

Weaknesses:

1. The authors point out that static-length rewards fail to account for problem difficulty. While the authors provide a Shapiro-Wilk test in Appendix C.1 to support this, reasoning lengths in LRMs can sometimes be multi-modal. If the distribution deviates significantly from Gaussian, the estimation of might be inaccurate.

2. The estimation of μ2 (mean length of correct responses) requires the model to generate correct answers during the rollout. for extremely difficult problems where the base model has near-zero accuracy (Pass@K≈0), the estimation of the optimal length becomes unstable or impossible. While the paper mentions clipping mechanisms, the behavior of the algorithm in "cold start" scenarios for very hard tasks could be discussed more thoroughly.

3. How is the 16.6% in the Abstract calculated? Which model's improvement does it refer to? This point is a bit confusing.

---

> ### Author Rebuttal · Authors · 2026-03-30
>
> Thank you for your insightful comments. We respond to your concerns as follows:
>
> ## Weakness 1
>
> We have supplemented the Hartigan's dip test for remaining approximately 40% of samples that did not pass the normal distribution, and the results are as follows:
>
> |  | Model | Unimodal Percentage |
> | --- | --- | --- |
> | All | DeepSeek-R1-Distill-1.5B | 89.9% |
> | All | SmarThinker-Distill-1.5B | 94.5% |
> | Correct Only | DeepSeek-R1-Distill-1.5B | 98.0% |
> | Correct Only | SmarThinker-Distill-1.5B | 100% |
>
> This indicates that the vast majority of length distributions do not exhibit multimodal properties. Length distributions that failed the Gaussian test usually did so because of a relatively uniform distribution or peak shift (due to the existence of a long tail). In such cases, using the mean and variance to describe these distributions remains effective, and we believe that the optimal length derived based on the Gaussian distribution assumption is still valid to a certain extent for these distributions.
>
> ## Weakness 2
>
> We agree that estimation becomes unstable when correct trajectories are scarce. However, our reward design mitigates this issue to a certain extent:
>
> 1. In extreme cases where there is at most one correct trajectory within a batch, the reward obtained by our algorithm is completely equivalent to that of standard GRPO; therefore, its performance is comparable to standard GRPO in this scenario.
> 2. If there are a small number of correct trajectories (more than one) in a batch, since we only apply length rewards to correct trajectories, the length reward for incorrect trajectories is 0 regardless of their output length. In this case, even if the length reward estimate is inaccurate, it only affects the allocation of importance among correct trajectories. Its effect approximates static length reward calculation based on a monotonic reward function.
>
> In summary, in extreme cases, our method would only degenerate to standard GRPO or a monotonic length reward scheme. These methods have been extensively verified and used, so the imprecise estimation of optimal length when correct trajectories are scarce does not bring about negative impacts.
>
> The issue you raised is very insightful, and we will add it to the paper.
>
> ## Weakness 3
>
> We apologize for the ambiguity. The 16.6% refers to the relative improvement of the DeepSeek-R1-7B model, increasing from 35.0% to 40.8%. We will clarify it in the new version.

---

### Official Review · Reviewer_Ypem · 2026-03-07

**Soundness:** 2
**Presentation:** 3
**Significance:** 2
**Originality:** 2
**Overall Recommendation:** 3
**Confidence:** 4

**Summary:**

This paper proposes a GRPO-based efficient reasoning method named SmartThinker. The proposed SmartThinker formulates the relationship between reasoning length and correctness using a Gaussian distribution to dynamically estimate an optimal reasoning length. Based on this, it guides over-long responses toward the optimal length and modulates the length reward coefficient to avoid penalizing correct reasoning paths. Experiments on mathematical benchmarks demonstrate that SmartThinker achieves length compression and improves the accuracy of reasoning models.

**Compliance With Llm Reviewing Policy:**

Affirmed.

**Key Questions For Authors:**

Please refer to Strengths And Weaknesses

**Limitations:**

yes

**Strengths And Weaknesses:**

**Strengths**

1. This paper proposes a dynamic optimal length estimation and a dynamic length reward coefficient , which dynamically adapt according to the relative problem difficulty without relying on static, blind linear length penalties.


2. The analysis of the overthinking phenomenon brings new insights into efficient LLM reasoning, i.e., the relationship between reasoning length and accuracy typically follows a U-shaped curve, and extending reasoning beyond an optimal point can harm accuracy.



**Weaknesses**

1. The theoretical analysis of the optimal reasoning length assumes that both the general length distribution and the successful length distribution follow Gaussian profiles. However, the authors may want to explain the statistical robustness of this assumption during training, given that the GRPO group size is only set to 8. Estimating stable Gaussian parameters (especially for the subset of correct trajectories) from such a small sample size might be unreliable.



2. Some concerns about the model scale are as follows. The authors evaluate the method on relatively small distilled models (e.g., 1.5B, 4B, and 7B). I suggest the authors consider discussing or evaluating the inductive setting of this method on much larger reasoning models (e.g., 32B+ parameters), as larger models might exhibit different "overthinking" dynamics and internal length-control capabilities.


3. The authors may want to clarify the baseline settings. Directly using open-sourced models for post-trained baselines instead of re-training them on the same dataset makes the comparison in Table 1 unconvincing.

4. Some concerns about training efficiency exist. Although SmartThinker requires fewer RL steps, dynamic computations introduce per-step complexity. The missing of wall-clock training time makes it difficult to assess the actual computational cost.

5. Minor Comments:

(1) The authors mention omitting the KL loss to improve training efficiency. I am a bit curious about this decision, as the KL penalty is typically an important component in RL fine-tuning to prevent policy degradation or collapse. Can it simply be removed without any negative impact on the model's general capabilities?

(2) There are instances of the word "Gaussion" in the manuscript (e.g., in the title of Table 5). This appears to be a typo and should probably be corrected to "Gaussian".

---

> ### Author Rebuttal · Authors · 2026-03-30
>
> Thank you for your insightful comments. We respond to your concerns as follows:
>
> ## Weakness 1
>
> We agree that estimating optimal length with a group size of 8 introduces variance. In practice, using a group size of 16 or 32 can yield more accurate estimates of the optimal length. Considering that our method requires fewer training steps, appropriately increasing the group size is acceptable. Moreover, since most baselines in our paper adopt a group size of 8 for training, we employ the same setting to ensure experimental fairness.
>
> We further quantify estimation error by comparing against a reference of 128 samples:
>
> | Model | Group Size | Median Rel. Err. |
> | --- | --- | --- |
> | Distill-1.5B | 8 | 10.5% |
> | Distill-1.5B | 16 | 8.7% |
> | Distill-1.5B | 32 | 7.6% |
>
> The results show that even when sampling only 8 trajectories, the median relative error is only around 10%. Compared to the reduction in transition length brought by the static length reward, the error of SmartThinker is acceptable.
>
> ## Weakness 2
>
> Due to computational constraints, we did not evaluate models beyond 7B. However, we believe our method has already demonstrated sufficient potential. On one hand, the baseline methods only trained 1.5B and 7B models. Direct comparisons at these two scales demonstrate the superiority of our method over others. On the other hand, prior work has discussed the relationship between output precision and length across different model scales. Specifically, please refer to the paper "When More is Less: Understanding Chain-of-Thought Length in LLMs". This paper conducted experiments on 1.5B, 7B, 32B, and 72B models, verifying that optimal length generally exists across models of different scales. Furthermore, the paper found that the relationship between precision and output length exhibits an inverted U-shape, which aligns with the hypothesis in our paper. Therefore, we believe the effectiveness of our method on small-scale models can be transferred to larger-scale models.
>
> In the future, we will attempt experiments on larger-scale models.
>
> ## Weakness 3
>
> We acknowledge this concern and clarify our choices:
>
> 1. The training datasets selected for different baselines are completely identical or originate from the same source. Among the baselines involved in the experiment, SmartThinker, LASER, and AutoThink are all trained using DeepScaleR-Preview-Dataset, while the training set used by ThinkPrune shares the same data source as DeepScaleR-Preview-Dataset.
> 2. In fact, we did reproduce some baselines but failed to achieve results identical to the open-source models, presumably due to subtle differences in hyperparameter settings. Considering that most open-source models undergo careful tuning of training settings and hyperparameters to achieve optimal performance, directly using open-source models avoids situations where baselines perform poorly due to parameter selection.
> 3. We conduct a new and detailed evaluation for all baselines. During this process, we discovered that most baselines reported lower accuracy in their original papers. The main reason for this may be that the mathematical verifiers used performed only simple string matching, rather than using complex verifiers such as math_verify. In this regard, we used a unified verifier in the evaluation to correct the experimental results. We believe that through a complete and comprehensive evaluation, our experimental results have been made as fair as possible.
>
> ## Weakness 4
>
> Dynamic optimal length computation involves only simple operations over small arrays (group size = 8), having almost no impact on the overall training time. Its impact is far less than that caused by the randomness during the rollout process.
>
> We test the training efficiency of the 7B model with length reward (i.e., SmartThinker) and without length reward (i.e., standard GRPO) under the same hardware conditions, environment, and training settings. We train both reward settings for the first 25 steps and calculate their evaluation time. The results show that standard GRPO takes an average of 116 seconds per step, and SmartThinker takes 114 seconds per step. This indicates that the impact of output length on training efficiency is far higher than the dynamic calculation of the length reward.
>
> ## Minor Comments
>
> (1) We removed KL divergence in our training because we observed that it does not have a significant impact when training for fewer than 150 steps. Not calculating the KL divergence eliminates the overhead of computing the reference model's log probability. Therefore, we did not use KL divergence in our experiments solely to accelerate training for rapid verification of results. In actual engineering practice, when SmartThinker is integrated into the post-training process, it is still recommended to include KL divergence to stabilize training.
>
> (2) We are sorry for the mistake. We will correct it in the new version.

---

> > ### Author Rebuttal · Reviewer_Ypem · 2026-04-04
> >
> > Thank you for the detailed rebuttal. Since some of my concerns remain unresolved, such as the evaluation on larger models, I will maintain my current score.

---

> > > ### Author Response · Authors · 2026-04-07
> > >
> > > Thank you for your feedback!
> > >
> > > To address your concerns, we have attempted to test the SmartThinker method using a 32B model. Our base model is DeepSeek-R1-Distill-Qwen-32B, and the training dataset and training parameters are consistent with those in the paper. We trained the model for 35 steps, and the results are as follows:
> > >
> > > |  | Math500 | | AIME25 | | AMC23 | | Mean | |
> > > | --- | --- | --- | --- | --- | --- | --- | --- | --- |
> > > |  | Acc | Len | Acc | Len | Acc | Len | Acc | Len |
> > > | Base Model | 95.5 | 3676 |  55.0 | 13587 | 95.0 | 5944 | 81.8 | 7736 |
> > > | SmartThinker | 95.1 | 3469 | 59.2 | 11482 | 97.5 | 5438 | 83.9 | 6796 |
> > >
> > > From the experimental results, it can be seen that our method can also achieve compressing the CoT length while improving model accuracy on the 32B model.
> > >
> > > Due to limited time, we were unable to perform fine-grained parameter tuning for the 32B model, as models of different scales require different learning rates and training steps. Moreover, since the DeepScaleR-Preview dataset is relatively easy for the 32B model, its learning effect also differs slightly from that of smaller-scale models.
> > >
> > > Hope the above results can resolve your concerns.

---

### Official Review · Reviewer_N8Do · 2026-03-08

**Soundness:** 3
**Presentation:** 3
**Significance:** 2
**Originality:** 3
**Overall Recommendation:** 4
**Confidence:** 4

**Summary:**

The paper proposes SmartThinker, an efficient reasoning method for Large Reasoning Models (LRMs) that mitigates the "overthinking" problem during reinforcement learning (specifically GRPO). Instead of relying on static length penalties or expensive Process Reward Models (PRMs), the authors estimate an "optimal reasoning length" by modeling the length distribution of correct and overall responses using a Gaussian assumption. Furthermore, the paper introduces a dynamic reward coefficient (Λ) to ensure that the normalized advantage of correct reasoning trajectories remains non-negative. This progressive length calibration achieves significant token reduction while maintaining or improving accuracy on reasoning tasks.

**Compliance With Llm Reviewing Policy:**

Affirmed.

**Final Justification:**

The rebuttal adequately addressed my main concerns. The new AIME26 results clear up the data contamination risk, and the entropy analysis shows that reasoning diversity is preserved. While the Gaussian assumption for small group sizes isn't perfect, the dynamic reward acts as a sufficient safety net. Overall, it's a practical and solid engineering contribution. I will maintain my score of 4.

**Key Questions For Authors:**

1. To address potential concerns regarding data contamination from the DeepScaleR-preview training set, it would significantly strengthen the claims if the authors could evaluate the method on a completely blind, newly released test set (e.g., 2026 math competitions). How might the performance hold up in such a strictly uncontaminated setting?

2. Regarding the diversity of reasoning paths, it would be insightful to see whether the proposed method inadvertently reduces generation variety. Providing metrics such as entropy analysis of the generated CoTs before and after training would be highly appreciated and help clarify this aspect.

I'd like to raise my score if my concerns are addressed.

**Limitations:**

yes

**Strengths And Weaknesses:**

**Strengths**:

1. Cost-Effective Design: The method elegantly addresses the length-accuracy trade-off without the need for step-wise human annotations or external PRMs, offering a highly practical solution for compute-constrained alignment.

2. Robust Reward Formulation: The introduction of the dynamic coefficient Λ is a clever mathematical safeguard. It theoretically prevents the model from receiving negative advantages for correct but overly verbose answers, which stabilizes the RL training process and prevents reasoning degradation.

3. Rigorous Statistical Attitude: Unlike many empirical RL papers, the authors explicitly conduct hypothesis testing (Shapiro-Wilk and KS tests) on their Gaussian assumption, showing a commendable level of scientific rigor.

**Weaknesses**:

1. Risk of Data Contamination: The evaluation relies heavily on benchmarks like AIME25. Since the model was trained on the DeepScaleR-preview dataset, which aggregates various math competition problems, there is a critical risk of data contamination. It is unclear if the 16.6% improvement is due to enhanced reasoning paradigms or merely a better recall of memorized templates under the new reward structure.

2. Threat to Decoding Diversity (Mode Collapse): By penalizing trajectories that deviate from the "optimal length," the reward mechanism acts as an entropy-reduction process. This may inadvertently penalize unconventional but valid reasoning paths, leading to mode collapse and capping the model's capacity to solve extreme out-of-distribution complex problems.

3. Fragility of the Gaussian Assumption: The authors' own tests show that only ~60% of the distributions pass the KS test. For tasks where length distribution is heavily skewed or multimodal, the calculated L_opt could drift significantly, injecting noisy reward signals.

---

> ### Author Rebuttal · Authors · 2026-03-30
>
> Thank you for your insightful comments. We respond to your concerns as follows:
>
> ## Weakness 1
>
> We agree that data contamination is an important concern. While the DeepScaleR-Preview dataset is indeed close in domain to benchmarks such as AIME25 and AMC23, we believe its use is justified for the following reasons:
>
> 1. Several baselines, including ShorterBetter, LASER, and AutoThink, are also trained on the DeepScaleR-Preview dataset, indicating its standard use in efficient reasoning research.
> 2. The dataset only contains AIME problems up to 2023 and AMC problems up to 2022. Therefore, there is no theoretical overlap with AIME25 or AMC23.
> 3. Compared to prior methods, our approach uses fewer training steps and smaller batch sizes, which reduces the risk of overfitting to this domain.
> 4. We additionally provide out-of-domain (OOD) evaluations in the appendix. Our method consistently improves accuracy while reducing reasoning length. You can refer to our response to Reviewer 91Wj's first weakness
>
> ## Weakness 2
>
> We agree that improperly designed rewards may lead to mode collapse. To mitigate this, we introduce a dynamic coefficient to ensure that all correct trajectories maintain a positive advantage. As for incorrect trajectories, we did not assign length rewards, so their advantage is always an identical non-positive number. The final result of the algorithm is more similar to assigning different importance to correct trajectories based on length, rather than penalizing trajectories that deviate from the optimal length.
>
> Compared to static length rewards, our design is more diversity-preserving:
>
> - Trajectories shorter than the optimal length receive identical rewards.
> - Longer trajectories are still assigned positive (but smaller) advantages due to dynamic scaling.
>
> This avoids overly aggressive penalization and allows the model to retain useful reasoning patterns.
>
> Currently, a large number of GRPO-based variants have emerged to address the issue of training instability (such as DAPO, GSPO, SAPO, etc.). These algorithms are compatible with our method because our method only involves the design of the reward.
>
> We also acknowledge that some useful patterns in incorrect or overly long trajectories may not be rewarded. We believe that methods such as introducing Process Reward Models (PRM) should be used to supervise the process. These process supervision methods are orthogonal to our outcome supervision and can work together.
>
> ## Weakness 3
>
> We thank the reviewer for pointing this out. To better analyze the ~40% of cases failing the Gaussian test, we additionally conduct Hartigan’s dip test:
>
> |  | Model | Unimodal Percentage |
> | --- | --- | --- |
> | All | DeepSeek-R1-Distill-1.5B | 89.9% |
> | All | SmarThinker-Distill-1.5B | 94.5% |
> | Correct Only | DeepSeek-R1-Distill-1.5B | 98.0% |
> | Correct Only | SmarThinker-Distill-1.5B | 100% |
>
> This indicates that the vast majority of length distributions do not exhibit multimodal properties. Length distributions that failed the Gaussian test usually did so because of a relatively uniform distribution or peak shift (due to the existence of a long tail). In such cases, using the mean and variance to describe these distributions remains effective,  and the estimated optimal length remains a reasonable approximation.
>
> ## Key Question 1
>
> We further evaluate on AIME26 to address contamination concerns:
>
> |  | Distill-1.5B |  | Qwen3-4B-Thinking |  |
> | --- | --- | --- | --- | --- |
> |  | Acc | len | Acc | len |
> | Base Model | 15.8% | 17232 | 73.3% | 20992 |
> | SmartThinker | 18.3% | 9989 | 71.7% | 13833 |
>
> From the experimental results, our method still brings accuracy improvements on AIME26 for the DeepSeek-Distill-1.5B model, while there is a slight accuracy decline on Qwen3-4B-Thinking. However, considering the substantial length compression, this magnitude of decline is acceptable.
> This also suggests that our method is more suitable for models like the DeepSeek-Distill series that have not undergone extensive post-training. For models like Qwen3-4B-Thinking-2507, since they have already been extensively fine-tuned for mathematical reasoning problems, it is relatively difficult to further improve their reasoning accuracy.
>
> ## Key Question 2
>
> We tested the entropy changes on AIME26 using 1.5B and 4B models. The results show that the entropy of the 1.5B model decreased from 0.68 to 0.43, and the 4B model decreased from 0.34 to 0.18. This level of entropy reduction is typical in RL training and remains within a reasonable range, indicating no severe diversity collapse.

---

> > ### Author Rebuttal · Reviewer_N8Do · 2026-04-01
> >
> > The authors' rebuttal, including AIME26 results and entropy analysis, has sufficiently addressed my concerns. I am satisfied and will maintain my score of 4.

---

> > > ### Author Response · Authors · 2026-04-07
> > >
> > > Thank you for your feedback!
> > >
> > > We have added experiments on the 32B model in our response to Reviewer Ypem's rebuttal comment. In addition, we have also tested the performance of this model on AIME26 and observed that the accuracy can be improved from 61.7% to 63.3%. This demonstrates that our method can still bring accuracy improvements on larger models and the latest benchmark.
> > >
> > > We hope our reply is helpful for your understanding of our work!

---

### Official Review · Reviewer_91Wj · 2026-03-13

**Soundness:** 3
**Presentation:** 2
**Significance:** 2
**Originality:** 2
**Overall Recommendation:** 3
**Confidence:** 3

**Summary:**

The paper proposes SmartThinker, a GRPO-based training framework for improving the efficiency of chain-of-thought reasoning by learning to control reasoning length. The authors observe that large reasoning models often produce unnecessarily long reasoning traces ("overthinking"), which increases computation and can even harm accuracy. SmartThinker addresses this by estimating the relationship between reasoning length and correctness from groups of sampled trajectories during training and identifying an optimal reasoning length that maximizes the probability of producing a correct answer. The training objective then guides overly long but correct reasoning trajectories toward this optimal length while dynamically adjusting the length reward coefficient to avoid penalizing valid reasoning paths. Experiments on mathematical reasoning benchmarks show that the method can reduce reasoning token usage by over 50% while maintaining or even improving accuracy, suggesting that efficient reasoning depends not only on generating explanations but also on learning when shorter reasoning processes are sufficient.

**Compliance With Llm Reviewing Policy:**

Affirmed.

**Key Questions For Authors:**

See my concerns in the weaknesses above.

Also, since the method relies primarily on outcome-based rewards (correct vs. incorrect) rather than process supervision, how well can it improve the quality of intermediate reasoning steps rather than only controlling reasoning length?

**Limitations:**

yes

**Strengths And Weaknesses:**

Strengths:

- The paper studies how to reduce unnecessary chain-of-thought tokens while maintaining reasoning accuracy, which targets a key cost bottleneck in reasoning-heavy applications.

- The proposed progressive calibration procedure provides a principled mechanism for aligning chain-of-thought length with the level of reasoning required for each problem.

- The analysis shows that excessively long chains of thought can harm performance, which motivates the need for explicit length calibration during training.

- Experiments indicate that the method reduces reasoning token usage while maintaining competitive performance on standard reasoning benchmarks.

- Since the method operates at the level of reasoning trajectory length rather than model architecture, it can potentially be integrated with a wide range of LLM training pipelines.

Weaknesses:

- While the method reduces reasoning tokens, the reported improvements in accuracy are generally small across benchmarks, which raises questions about whether the additional calibration procedure is justified.

- The approach assumes that an "optimal" chain-of-thought length exists for each problem. The paper provides limited theoretical justification for this assumption, and it is unclear whether such an optimal length is stable across models, datasets, or prompting formats.

- The experiments do not explore more complex multi-step tasks or real-world settings where reasoning trajectories may be less structured.

- The progressive calibration procedure introduces additional tuning choices, but the paper provides limited ablation on how these choices affect performance.

- Shorter reasoning traces may reduce token usage, but the paper provides limited evidence that the approach improves robustness or generalization beyond efficiency gains.

---

> ### Author Rebuttal · Authors · 2026-03-30
>
> Thank you for your insightful comments. We respond to your concerns as follows:
>
> ## Weakness 1
>
> We respectfully argue that the additional length calibration is justified for the following reasons:
>
> 1. Our primary objective is to minimize reasoning length without sacrificing accuracy. On this basis, we observe that our method enables a synergistic improvement in both efficiency and accuracy, which is a non-trivial and surprising result.
> 2. Only our method and ThinkPrune improve average accuracy, but ThinkPrune relies heavily on AMC23 while degrading elsewhere. Our method achieves consistent gains across benchmarks and scales, moving beyond the typical length-accuracy trade-off to joint optimization.
> 3. Compared with post-training processes requiring large-scale data, our method needs only a few dozen additional training steps with minimal computational overhead.
>
> ## Weakness 2
>
> Prior work has already established the existence of an optimal reasoning length in LLMs. Our method builds upon these findings by introducing a dynamic estimation mechanism and applying it to efficient reasoning. The paper "When More is Less: Understanding Chain-of-Thought Length in LLMs" tests Qwen2.5 (1.5B~72B) and Llama3.1 (8B, 70B) on different datasets, verifying that optimal length universally exists across models of different architectures and scales, as well as datasets of different domains and difficulties.
>
> ## Weakness 3
>
> We focus on mathematical reasoning during training for the following reasons:
>
> 1. All baselines are trained on math datasets. Specifically, ShorterBetter, LASER, and AutoThink use DeepScaleR-Preview, while ThinkPrune relies on a closely related synthetic dataset.
> 2. Beyond math, we evaluate on code generation tasks (Appendix), where our method achieves both accuracy gains and length reduction.
> 3. For tasks such as knowledge QA, CoT is typically short and does not benefit from specialized training.
>
> The results for the OOD tasks in the appendix are shown below:
>
> |  | MMLU |  | MathQA |  | LiveCode |  | Humaneval |  |
> | --- | --- | --- | --- | --- | --- | --- | --- | --- |
> |  | Acc | Len | Acc | Len | Acc | Len | Acc | Len  |
> | Distill-1.5B | 46.02 | 1686 | 84.0 | 5206 | 24.2 | 10668 | 68.9 | 4740 |
> | SmartThinker-Distill-1.5B | 46.48 | 994 | 84.6 | 2581| 25.4 | 7658 | 69.5 | 3099 |
>
> We have additionally conducted experiments on the 4B model:
>
> |  | MMLU |  | MathQA |  | LiveCode |  | Humaneval |  |
> | --- | --- | --- | --- | --- | --- | --- | --- | --- |
> |  | Acc | Len | Acc | Len | Acc | Len | Acc | Len  |
> | Qwen3-4B-Thinking | 80.8 | 2978 | 91.3 | 3253 | 68.1 | 12388 | 95.7 | 4507 |
> | SmartThinker-Qwen3-4B | 81.1 | 2167 | 91.7 | 2224 | 68.4 | 9376 | 96.5 | 3158 |
>
> ## Weakness 4
>
> If “tuning choices” refers to hyperparameters, we would like to clarify that our method introduces no additional hyperparameters. Both the optimal length and reward coefficients are computed adaptively based on the proposed algorithm. This is a key advantage, as it eliminates the need for manual tuning.
>
> ## Weakness 5
>
> Our goal is to compress the length of CoT as much as possible while ensuring that the model's accuracy and generalization capability are not compromised. Therefore, as long as the model's generalization capability is not compromised, the compression method is effective. In the appendix, we have demonstrated that the 1.5B model can also synergistically optimize inference accuracy and efficiency in OOD scenarios. We have also added experiments with the 4B model, as shown in our answer to the third question. The result shows that training on math problems can also lead to improvement in both accuracy and efficiency on problems in other domains. This demonstrates that our method enhances the model's **generalization capability**.
>
> ## Key Question 1
>
> A shorter chain-of-thought length implies higher information density, and the model reduces the number of tokens wasted on exploring meaning trajectories. Furthermore, our method shortens the length using a relatively conservative strategy. This enables the model to retain the capability to explore different paths.
> To quantify the optimization effect of the model's chain-of-thought structure, we have additionally used LLM-as-judge to classify the model's reasoning structure. We use DeepSeek-V3.2 for discrimination and employ the same system prompt as in ShorterBetter‘s paper. We examine the reasoning structures of Distill-1.5B on 50 randomly selected problems from math500 before and after training:
>
> |  | Pivotal Reasoning | Repetition & Rephrasing | Exploring Alternatives | Verification & Explanation | Non-Substantive Statement |
> | --- | --- | --- | --- | --- | --- |
> | Distill-1.5B | 6.13% | 1.13% | 86.86% | 2.73% | 3.05% |
> | SmartThinker-Distill-1.5B | 16.04% | 0.68% | 77.29% | 2.90% | 3.09% |
>
> It shows that after training, the model reduces the proportion of erroneous path exploration and information repetition, while increasing the proportion of key reasoning paths.

---

> > ### Author Rebuttal · Reviewer_91Wj · 2026-04-03
> >
> > I want to thank the authors for discussions and additional results. I will keep my original score unchanged.

---

> > > ### Author Response · Authors · 2026-04-07
> > >
> > > Thank you for your feedback! Your comments are valuable to us.
> > >
> > > We have added experiments on the 32B model in our response to Reviewer Ypem‘s rebuttal comment. We hope this new content is helpful for your understanding of our work.

---

### Decision · Program_Chairs · 2026-04-30

**Decision:**

Accept (regular)

**Comment:**

The paper introduces SmartThinker, a GRPO-based training framework designed to mitigate "overthinking" in LRMs. The core contribution is a dynamic length calibration mechanism that estimates an "optimal" reasoning length for a given problem—modeled via Gaussian distributions—and applies rewards to steer correct trajectories toward this length. Unlike static length penalties, SmartThinker adaptively adjusts the penalty based on problem difficulty and current model performance.

Main Strengths:
- The method demonstrates significant token reduction (up to 50%) while maintaining or slightly improving accuracy, which is a critical practical concern for deploying LRMs.
- Reviewers appreciated moving away from heuristic, one-size-fits-all length penalties toward a more principled, data-driven reward structure.
- The inclusion of dynamic coefficients ($\Lambda$) to prevent negative advantages for correct answers and the statistical testing of the Gaussian assumption were noted as strengths.

Main Weaknesses and Rebuttal Discussion:

The reviews were initially mixed (two Weak Rejects, two Weak Accepts). The discussion focused on several key points:

- Reviewers (91Wj, N8Do, Ypem) questioned whether reasoning lengths are truly Gaussian. The authors provided Hartigan’s dip tests during the rebuttal, showing high unimodality percentages (>90%), which largely satisfied Reviewer N8Do.

- Concerns were raised regarding whether improvements were due to data contamination (Reviewer N8Do). The authors countered with new results on AIME26 (a "blind" test set relative to their training data) and OOD benchmarks (MMLU, HumanEval), showing sustained efficiency and accuracy gains.

- Reviewer Ypem remained concerned that the method was only tested on smaller models (up to 7B). While the authors cited external literature suggesting the "inverted U-shape" of length vs. accuracy holds for larger models, the lack of 32B+ experiments remains a limitation.

- Reviewer 91Wj felt the accuracy gains were marginal. The authors clarified that the primary goal is efficiency (token reduction) without accuracy degradation; in that context, any accuracy "gain" is a bonus.

On one hand, the absolute accuracy improvements are modest, and the evaluation is limited to smaller-scale models. On the other hand, the efficiency gains are substantial and well-validated across multiple benchmarks (including a newly released test set during rebuttal).